# Tensor Programs V:
# Tuning Large Neural Networks via
# Zero-Shot Hyperparameter Transfer

**Greg Yang**[*×]  **Edward J. Hu**[*×†]  **Igor Babuschkin**[°]  **Szymon Sidor**[°]  **Xiaodong Liu**[×]
**David Farhi**[°]  **Nick Ryder**[°]  **Jakub Pachocki**[°]  **Weizhu Chen**[×]  **Jianfeng Gao**[×]
[×]Microsoft Corporation    [°]OpenAI

## Abstract

Hyperparameter (HP) tuning in deep learning is an expensive process, prohibitively so for neural networks (NNs) with billions of parameters. We show that, in the recently discovered Maximal Update Parametrization ($\mu$P), many optimal HPs remain stable even as model size changes. This leads to a new HP tuning paradigm we call *$\mu$Transfer*: parametrize the target model in $\mu$P, tune the HP indirectly on a smaller model, and *zero-shot transfer* them to the full-sized model, i.e., without directly tuning the latter at all. We verify $\mu$Transfer on Transformer and ResNet. For example, 1) by transferring pretraining HPs from a model of 13M parameters, we outperform published numbers of BERT-large (350M parameters), with a total tuning cost equivalent to pretraining BERT-large once; 2) by transferring from 40M parameters, we outperform published numbers of the 6.7B GPT-3 model, with tuning cost only 7% of total pretraining cost. A Pytorch implementation of our technique can be found at `github.com/microsoft/mup`.[2]

## 1  Introduction

Hyperparameter (HP) tuning is critical to deep learning. Poorly chosen HPs result in subpar performance and training instability. Many published baselines are hard to compare to one another due to varying degrees of HP tuning. These issues are exacerbated when training extremely large deep learning models, since state-of-the-art networks with billions of parameters become prohibitively expensive to tune.

Recently, [45] showed that different neural network parametrizations induce different infinite-width limits and proposed the *Maximal Update Parametrization (abbreviated $\mu$P)* (summarized in Table 3) that enables "maximal" feature learning in the limit. Intuitively, it ensures that each layer is updated on the same order during training *regardless of width*.[3] In contrast, while the standard parametrization (SP) ensures activations are of unit order at initialization, it actually causes

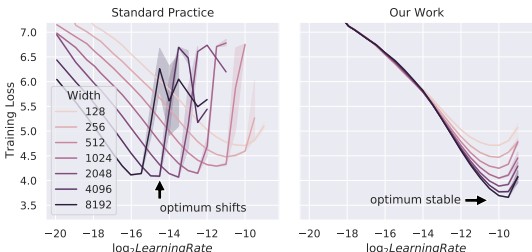

Figure 1: Training loss against learning rate on Transformers of varying $d_{model}$ trained with Adam. Conventionally and in contrast with our technique, different widths do not share the same optimal hyperparameter; wider networks do not always perform better than narrower ones; in fact they underperform the same-width networks in our technique even after tuning learning rate. See Sections 3 and 4 for experimental setup.

---

[†]Work done partly during Microsoft AI Residency Program.
[*]Equal contribution. Order is random. Correspondence to {gregyang, edwardhu}@microsoft.com
[2]See `arxiv.org` for the full, up-to-date version of this work.
[3]i.e., the updates' effect on activations becomes roughly independent of width in the large width limit.

35th Conference on Neural Information Processing Systems (NeurIPS 2021).

---
**Algorithm 1** Tuning a Large Target Model via $\mu$Transfer
---
1: Parametrize target model in Maximal Update Parametrization ($\mu$P)
2: Tune a smaller version (in width and/or depth) of target model
3: Copy tuned hyperparameters to target model
---

Table 1: **Hyperparameters That Can Be $\mu$Transferred, Not $\mu$Transferred, or $\mu$Transferred Across,** with a few caveats discussed in Section 5.1. * means *empirically validated only* on Transformers, while all others additionally have theoretical justification.

| $\mu$Transferable | Not $\mu$Transferable | $\mu$Transferred *Across* |
|---|---|---|
| optimization related, init, parameter multipliers, etc | regularization (dropout, weight decay, etc) | width, depth*, batch size*, training time*, seq length* |

them to blow up in wide models during training [45] essentially due to an imbalance of per-layer learning rate (also see Fig. 8). We leverage $\mu$P to *zero-shot transfer HPs from small models to large models* in this work – that is, we obtain near optimal HPs on a large model without directly tuning it at all! While practitioners have always guessed HPs of large models from those of small models, the results are hit-or-miss at best because of incorrect parametrization. For example, as shown in Fig. 1, in a Transformer, the optimal learning rate is stable with width in $\mu$P (right) but far from so in standard parametrization (left). In addition to width, we empirically verify that, with a few caveats, HPs can also be transferred across depth (in Section 5.1) as well as batch size, language model sequence length, and training time (in Appendix I.2.1). This reduces the tuning problem of an (arbitrarily) large model to that of a (fixed-sized) small model. Our overall procedure, which we call *$\mu$Transfer*, is summarized in Algorithm 1 and Fig. 2, and the HPs we cover are summarized in Tables 1 and 2.

There are several benefits to our approach: 1. **Better Performance:** $\mu$Transfer is not just about predicting how the optimal learning rate scales in SP. In general, we expect the $\mu$Transferred model to outperform its SP counterpart with learning rate optimally tuned. For example, this is the case in Fig. 1 with the width-8192 Transformer. We discuss the reason for this in Appendices B and C. 2. **Speedup:** It provides massive speedup to the tuning of large models. For example, we are able to outperform published numbers of (350M) BERT-large [9] purely by zero-shot HP transfer, with tuning cost approximately equal to 1 BERT-large pretraining. Likewise, we outperform the published numbers of the 6.7B GPT-3 model [6] with tuning cost being only 7% of total pretraining cost. For models on this scale, HP tuning is not feasible at all without our approach. 3. **Tune**

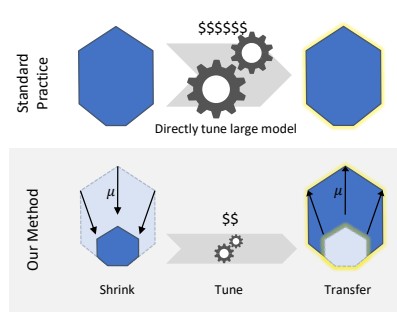

Figure 2: Illustration of $\mu$Transfer

**Once for Whole Family:** For any fixed family of models with varying width and depth (such as the BERT family or the GPT-3 family), we only need to tune a single small model and can reuse its HPs for all models in the family.[4] For example, we will use this technique to tune BERT-base (110M parameters) and BERT-large (350M parameters) simultaneously by transferring from a 13M model. 4. **Better Compute Utilization:** While large model training needs to be distributed across many GPUs, the small model tuning can happen on individual GPUs, greatly increasing the level of parallelism for tuning (and in the context of organizational compute clusters, better scheduling and utilization ratio). 5. **Painless Transition from Exploration to Scaling Up:** Often, researchers explore new ideas on small models but, when scaling up, find their HPs optimized during exploration work poorly on large models. $\mu$Transfer would solve this problem.

Nevertheless, $\mu$Transfer still has several limitations. For example, while it is very effective for pretraining, it cannot transfer regularization HPs,[5] so it's generally not applicable to the finetuning of pretrained models. We discuss other limitations carefully in Section 5.1.

---

[4]but possibly *not* for different data and/or tasks.

[5]It can transfer regularization HPs to the extent they help *training* but it may not transfer their effect on *testing*.

Table 2: **Examples of $\mu$Transferable Hyperparameters.** All of the below can also be specialized to per-layer hyperparameters.

| Optimizer Related | Initialization | Parameter Multipliers |
|:---:|:---:|:---:|
| learning rate (LR), momentum, Adam beta, LR schedule, etc | per-layer init. variance | multiplicative constants after weight/biases, etc |

**Our Contributions**

- We demonstrate it is possible to zero-shot transfer near optimal HPs to a large model from a small version via the Maximal Update Parametrization ($\mu$P) from [45].
- While [45] only covered SGD, here we derive $\mu$P for Adam as well (Table 3).
- We propose a new HP tuning technique, *$\mu$Transfer*, for large neural networks based on this observation that provides massive speedup over conventional methods and covers both SGD and Adam training;
- We thoroughly verify our method on machine translation and large language model pretraining (in Section 6.3) as well as image classification (in Appendix I.1);
- We release a PyTorch [27] package for implementing $\mu$Transfer painlessly. A sketch of this package is given in Appendix J.

**Terminologies**   Sometimes, to be less ambiguous, we often refer to the "large model" as the *target model*, as it is the model we wish to ultimately tune, while we refer to the "small model" as the *proxy model*, as it proxies the HP tuning process. We follow standard notation $d_{model}, d_{head} = d_k, d_v, n_{head}, d_{ffn}$ regarding dimensions in a Transformer; one can see Fig. 11 for a refresher.

***Tensor Programs* Series**   This paper is the 5th installment of the *Tensor Programs* series. While the target audience here are practitioners and empirical researchers, this paper presents the first major *practical* payoff of the *theoretical* foundation built in previous works [41–46].

## 2   Parametrization Matters: A Primer

In this section, we give a very basic primer on why the correct parametrization can allow HP transfer across width, but see Appendices L.1 to L.3 for more (mathematical) details.

The Central Limit Theorem (CLT) says that, if $x_1, \ldots, x_n$ are iid samples from a zero-mean, unit-variance distribution, then $\frac{1}{\sqrt{n}}(x_1 + \cdots + x_n)$ converges to a standard Gaussian $\mathcal{N}(0, 1)$ as $n \to \infty$. Therefore, we can say that $\frac{1}{\sqrt{n}}$ is the right order of *scaling factor* $c_n$ such that $c_n(x_1 + \cdots + x_n)$ converges to something nontrivial. In contrast, if we set $c_n = 1/n$, then $c_n(x_1 + \cdots + x_n) \to 0$; or if $c_n = 1$, then $c_n(x_1 + \cdots + x_n)$ blows up in variance as $n \to \infty$.

Now suppose we would like to minimize the function

$$F_n(c) \stackrel{\text{def}}{=} \mathop{\mathbb{E}}_{x_1, \ldots, x_n} f(c(x_1 + \cdots + x_n)) \tag{1}$$

over $c \in \mathbb{R}$, for some bounded continuous function $f : \mathbb{R} \to \mathbb{R}$. If we reparametrize $c = \alpha/\sqrt{n}$ for $\alpha \in \mathbb{R}$, then by CLT, $G_n(\alpha) \stackrel{\text{def}}{=} F_n(c) \to \mathbb{E} f(\mathcal{N}(0, \alpha^2))$ stabilizes into a function of $\alpha$ as $n \to \infty$. Then for sufficiently large $n$, the optimal $\alpha_n^* \stackrel{\text{def}}{=} \arg\min_\alpha G_n(\alpha)$ should be close to $\alpha_N^*$ for any $N > n$, and indeed, for $N = \infty$ — this precisely means we can *transfer* the optimal $c_n^*$ or $\alpha_n^*$ for a smaller problem (say $F_n$) to a larger problem (say $F_N$): $G_N$ is approximately minimized by $\alpha_n^*$ and $F_N$ is approximately minimized by $c_n^* \sqrt{n/N}$. Because the transfer algorithm is simply copying $\alpha$, we say the parametrization $c = \alpha/\sqrt{n}$ is the *correct parametrization* for this problem.

In the scenario studied in this paper, $x_1, \ldots, x_n$ are akin to randomly initialized parameters of a width-$n$ neural network, $c$ is akin to a HP such as learning rate, and $f$ is the test-set performance of the network *after training*, so that $F_n$ gives its expectation over random initializations. Just as in this example, if we parametrize the learning rate and other HPs correctly, then we can directly copy the optimal HPs for a narrower network into a wide network and expect approximately optimal

performance — this is the *hyperparameter transfer* we propose here. It turns out the Maximal Update Parametrization ($\mu$P) introduced in [45] is correct (akin to the parametrization in $\alpha$ above), while the standard parametrization (SP) is incorrect (akin to the parametrization in $c$). We will review both parametrizations shortly. Theoretically, a $\mu$P network has a well-defined infinite-width limit — akin to $(x_1 + \cdots + x_n)/\sqrt{n}$ having a $\mathcal{N}(0,1)$ limit by CLT — while a SP network does not (the limit will blow up) [45].[6] In fact, based on the theoretical foundation laid in [45], we argue in Appendix L.3 that $\mu$P should also be the *unique* parametrization that allows HP transfer across width.

We emphasize that, to ensure transferability of any hyperparameter (such as learning rate), it's not sufficient to reparametrize *only* that hyperparameter, but rather, we need to identify and correctly reparametrize *all* hyperparameters in Table 2. For example, in Fig. 1, the wide models in SP still underperform their counterparts in $\mu$P, even with learning rate tuned optimally. This is precisely because SP does not scale parameter multipliers and input/output layer learning rates correctly in contrast to $\mu$P (see Table 3). See Appendix C for more intuition via a continuation of our example here. We shall also explain this more concretely in the context of neural networks in Appendix B.

## 3 Hyperparameters Don't Transfer Conventionally

In the community there seem to be conflicting assumptions about HP stability. *A priori*, models of different sizes don't have any reason to share the optimal HPs. Indeed, papers aiming for state-of-the-art results often tune them separately. On the other hand, a nontrivial fraction of papers in deep learning fixes all HPs when comparing against baselines, which reflects an assumption that the optimal HPs should be stable — not only among the same model of different sizes but also among models of different designs — therefore, such comparisons are fair. Here, we demonstrate HP *instability* across width explicitly in MLP and Transformers in the standard parametrization. We will only look at training loss to exclude the effect of regularization.

**MLP with Standard Parametrization** We start with a 2-hidden-layer MLP with activation function $\phi$, using the standard parametrization[7] with LeCun initialization[8] akin to the default in PyTorch:

$$f(\xi) = W^{3\top}\phi(W^{2\top}\phi(W^{1\top}\xi + b^1) + b^2)$$
$$\text{with init.} \quad W^1 \sim \mathcal{N}(0, 1/d_{in}), \; W^{\{2,3\}} \sim \mathcal{N}(0, 1/n), \; b^{\{1,2\}} = 0, \tag{2}$$

where $W^1 \in \mathbb{R}^{d_{in} \times n}, b^1 \in \mathbb{R}^n$, $W^2 \in \mathbb{R}^{n \times n}, b^2 \in \mathbb{R}^n, W^3 \in \mathbb{R}^{n \times d_{out}}$ and $d_{in}$, $n$, and $d_{out}$ are the input, hidden, and output dimensions. The particular MLP we use has $\phi = ReLU$ and a cross-entropy (xent) loss function. We define the width of MLP as the hidden size $n$, which is varied from 256 to 8192. The models are trained on CIFAR-10 for 20 epochs, which is more than enough to ensure convergence.

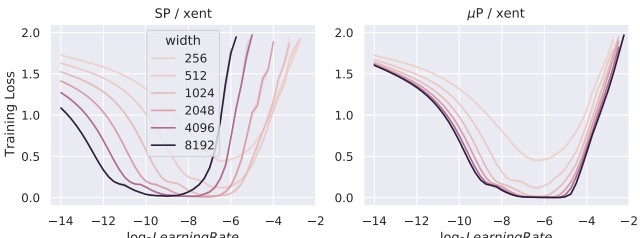

As shown on the left in Fig. 3, the optimal learning rate shifts by roughly an order of magnitude as the width increases from 256 to 8192; using the optimal learning of the smallest model on the largest model gives very bad performance, if not divergence.

Figure 3: MLP width different hidden sizes trained for 20 epoch on CIFAR-10 using SGD. **Left** uses standard parametrization (SP); **right** uses maximal update parametrization ($\mu$P). $\mu$P networks exhibit better learning rate stability than their SP counterparts.

**Transformer with Standard Parametrization** This perhaps unsurprising observation holds for more complex architectures such as Transformer as well, as shown in Fig. 1 (left). We define width

---

[6]The more theoretically astute reader may observe that SP with a $\Theta(1/width)$ learning rate induces a well-defined infinite-width limit exists as well. Nevertheless, this does not allow HP transfer because this limit is in kernel regime as shown in [45]. See Appendix L.3 for more discussions.

[7]i.e. the default parametrization offered by common deep learning frameworks. See Table 3 for a review.

[8]The key here is that the init. variance $\propto 1/\texttt{fan\_in}$, so the same insights here apply with e.g. He initialization.

Table 3: $\mu$**P[45] and SP for General Neural Networks, Basic Form.** This basic form emphasizes the *scaling with width (*fan_in *or* fan_out*)*; in practice, we may insert tunable multipliers in front of fan_in and fan_out as in Eq. (4). Notations: 1) $\eta$ is the "master" learning rate. 2) The fan_out of a bias vector is its dimension (whereas fan_in is 1). 3) Purple text highlights key differences from standard parametrization (SP); Gray text recalls the corresponding SP. *SGD* (resp. *Adam*) here can be replaced by variants such as SGD with momentum (resp. Adagrad, Adadelta, etc). In general, the three columns here can be interpreted as linear layers that have {finite, infinite, infinite} input dimension and {infinite, finite, infinite} output dimension in an infinite-width network; this description generalizes more readily to other parameters such as those of layernorm. Transformer $\mu$P requires one more modification ($1/d$ attention instead of $1/\sqrt{d}$); see Definition 4.1. This version of $\mu$P gets rid of parameter multipliers; for the version similar to that in [45], see Table 13. Also see Table 12 for a $\mu$P formulation that is easier to implement (and compatible with input/output weight sharing).

|  | Input weights & all biases | Output weights | | Hidden weights | |
|---|---|---|---|---|---|
| Init. Var. | $1/\text{fan\_in}$ | $1/\text{fan\_in}^2$ | $(1/\text{fan\_in})$ | $1/\text{fan\_in}$ | |
| SGD LR | $\eta \cdot \text{fan\_out}$ $\quad(\eta)$ | $\eta/\text{fan\_in}$ | $(\eta)$ | $\eta$ | |
| Adam LR | $\eta$ | $\eta/\text{fan\_in}$ | $(\eta)$ | $\eta/\text{fan\_in}$ | $(\eta)$ |

as $d_{model}$, with $d_k = d_q = d_v = d_{model}/n_{head}$ and $d_{ffn} = 4d_{model}$. The models are trained on wikitext-2 for 5 epochs. In Fig. 18 in the appendix we also show the instability of initialization scale and other HPs.

## 4  Unlocking Zero-Shot Hyperparameter Transfer with $\mu$P

We show that $\mu$P solves the problems we see in Section 3.

**MLP with $\mu$P**  For the MLP in Section 3, to switch to $\mu$P, we just need to modify Eq. (2)'s initialization of the last layer and its learning rates of the first and last layer as well as of the biases. The *basic form* is[9]

$$\text{initialize} \quad W^1 \sim \mathcal{N}(0, 1/d_{in}), \ W^2 \sim \mathcal{N}(0, 1/n), \ W^3 \sim \mathcal{N}(0, 1/n^2), \ b^{\{1,2\}} = 0$$
$$\text{with SGD learning rates} \quad \eta_{W^1} = \eta_{b^1} = \eta_{b^2} = \eta n, \ \eta_{W^2} = \eta, \ \eta_{W^3} = \eta n^{-1}. \tag{3}$$

Here, $\eta$ specifies the "master" learning rate, and we highlighted in purple the differences in the two parametrizations. This basic form makes clear the *scaling with width* $n$ of the parametrization, but in practice we will often insert (possibly tune-able) multiplicative constants in front of each appearance of $n$. For example, this is useful when we would like to be consistent with a SP MLP at a *base width* $n_0$. Then we may insert constants as follows: For $\tilde{n} \overset{\text{def}}{=} n/n_0$,

$$\text{initialize} \quad W^1 \sim \mathcal{N}(0, 1/d_{in}), \ W^2 \sim \mathcal{N}(0, 1/n), \ W^3 \sim \mathcal{N}(0, 1/n \cdot \tilde{n}), \ b^{\{1,2\}} = 0$$
$$\text{with SGD learning rates} \quad \eta_{W^1} = \eta_{b^1} = \eta_{b^2} = \eta \tilde{n}, \ \eta_{W^2} = \eta, \ \eta_{W^3} = \eta \tilde{n}^{-1}. \tag{4}$$

Then at width $n = n_0$, all purple factors above are 1, and the parametrization is identical to SP (Eq. (2)) at width $n_0$. Of course, as $n$ increases from $n_0$, then Eq. (4) quickly deviates from Eq. (2). In other words, for a particular $n$, $\mu$P and SP can be identical up to the choice of some constants (in this case $n_0$), but $\mu$P determines a different "set" of networks and optimization trajectory than SP as one varies $n$. As we will see empirically in the next section, this deviation is crucial for HP transfer.

Indeed, in Fig. 3(right), we plot the CIFAR10 performances, over various learning rates and widths, of $\mu$P MLPs with $n_0 = 128$. In contrast to SP, the optimal learning rate under $\mu$P is stable. This means that, the best learning rate for a width-128 network is also best for a width-8192 network in $\mu$P — i.e. HP transfer *works* — but not for SP. In addition, we observe performance for a fixed learning rate always weakly improves with width in $\mu$P, but not in SP.

This MLP $\mu$P example can be generalized easily to general neural networks trained under SGD or Adam, as summarized in Table 3, which is derived in Appendix L.

---

[9]While superficially different, this parametrization is equivalent to the $\mu$P defined in [45].

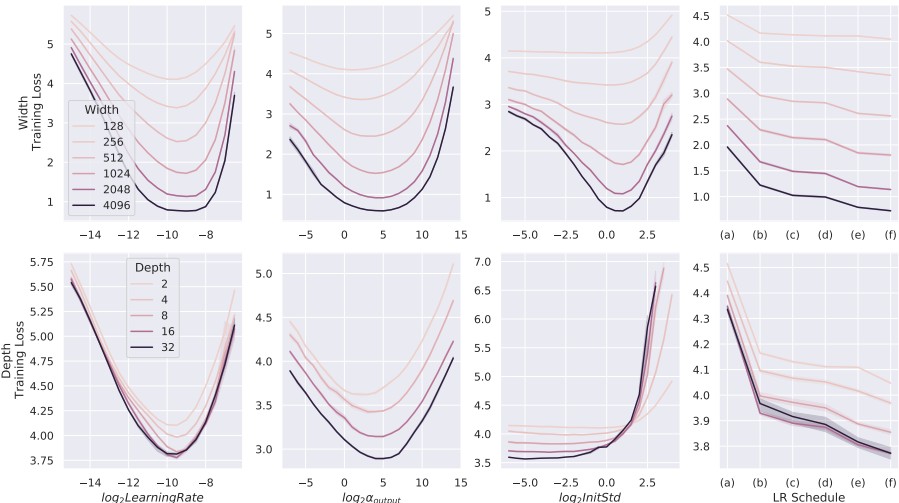

Figure 4: **Empirical validation of the stability of four representative hyperparameters on pre-LN Transformers in** $\mu$**P**: learning rate, last layer weight multiplier $\alpha_{output}$, weight initialization standard deviation, and learning rate schedule. We use the following learning rate schedules: (a) linear decay; (b) StepLR @ [5k, 8k] with a decay factor of 0.1; (c) StepLR @ [4k, 7k] with a decay factor of 0.3; (d) cosine annealing; (e) constant; (f) inverse square-root decay. All models are trained on wikitext-2 for 10k steps. When not specified in the legend, the width used is 256, depth 2, batch size 20, sequence length 256, and LR schedule constant. We sweep a particular HP, corresponding to each column, while fixing all others constant. See Section 5.1 for discussion of these results.

**Transformers with $\mu$P**    We repeat the experiments with base width $n_0 = 128$ for Transformers:

**Definition 4.1.** The *Maximal Update Parametrization ($\mu$P) for a Transformer* is given by Table 3 and $1/d$ attention instead of $1/\sqrt{d}$, i.e. the attention logit is calculated as $q^\top k/d$ instead of $q^\top k/\sqrt{d}$ where query $q$ and key $k$ have dimension $d$.[10]

The results are shown on the right in Fig. 1, where the optimal learning rate is stable, and the performance improves monotonically as width increases.

## 5    Which Hyperparameters Can Be $\mu$Transferred?

In this section, we explore how common HPs fit into our framework. In general, they can be divided into three kinds, summarized in Table 1:

1. those that can transfer from the small to the large model, such as learning rate (Table 2);

2. those that primarily control regularization and don't work well with our technique; and

3. those that define training *scale*, such as width as discussed above as well as others like depth and batch size, across which we transfer other HPs.

Those in the first category transfer across width, as theoretically justified above in Section 2. To push the practicality and generality of our technique, we empirically explore the transfer across the other dimensions in the third category. Note that $\mu$Transfer across width is quite general, e.g. it allows varying width ratio of different layers or number of attention heads in a Transformer; see Appendix G.2. This will be very useful in practice. For the second category, the amount of regularization (for the purpose of controlling overfitting) naturally depends on both the model size and data size, so we should not expect transfer to work if the parametrization only depends on model size. We discuss these HPs in more detail in Appendix G.1.

---

[10]This is roughly because during training, $q$ and $k$ will be correlated so $q^\top k$ actually scales like $d$ due to Law of Large Numbers, in contrast to the original motivation that $q$, $k$ are uncorrelated at initialization so Central Limit applies instead. See Appendix L.2.1 for a more in-depth discussion.

## 5.1 Empirical Validation and Limitations

Our empirical investigations focus on Transformers (here) and ResNet (in Appendix I.1.1), the most popular backbones of deep learning models today. We train a 2-layer pre-layernorm $\mu$P[11] Transformer with 4 attention heads on Wikitext-2. We sweep one of four HPs (learning rate, output weight multiplier, initialization standard deviation, and learning rate schedule) while fixing the others and sweeping along width and depth (with additional results in Fig. 19 on transfer across batch size, sequence length, and training time). Fig. 4 shows the results averaged over 5 random seeds.

Empirically, we find that for language modeling on Transformers, HPs generally transfer across scale dimensions if some minimum width (e.g. 256), depth (e.g., 4), batch size (e.g., 32), sequence length (e.g., 128), and training steps (e.g., 5000) are met, with some caveats discussed below. While the exact optimum can shift slightly with increasing scale, this shift usually has very small impact on the loss, compared to SP (Figs. 1 and 3(left)). However, there are some caveats. For example, the best initialization standard deviation does not seem to transfer well across depth (2nd row, 3rd column), despite having a stabler optimum across width. In addition, while our results on width, batch size, sequence length, and training time still hold for post-layernorm (Fig. 17),[12] the transfer across depth only works for pre-layernorm Transformer. Nevertheless, in practice (e.g. our results in Section 6.3) we find that fixing initialization standard deviation while tuning other HPs works well when transferring across depth.

# 6 Efficiency and Performance of $\mu$Transfer

Now that the plausibility of $\mu$Transfer has been established in toy settings, we turn to more realistic scenarios to see if one can achieve tangible gains. Specifically, we perform HP tuning only on a smaller proxy model, test the obtained HPs on the large target model directly, and compare against baselines tuned using the target model. We seek to answer the question: Can $\mu$Transfer make HP tuning more efficient while achieving performance on par with traditional tuning? As we shall see by the end of the section, the answer is positive. We focus on Transformers here, while experiments on ResNets on CIFAR10 and Imagenet can be found as well in Appendix I.1. All of our experiments are run on V100 GPUs.

## 6.1 Transformer on IWSLT14 De-En

**Setup** IWSLT14 De-En is a well-known machine translation benchmark. We use the default IWSLT (post-layernorm) Transformer implemented in `fairseq` [25] with 40M parameters, which we denote as the *1x model*.[13] For $\mu$Transfer, we tune on a *0.25x model* with $1/4$ of the width, amounting to 4M parameters. For this experiment, we tune via random search the learning rate $\eta$, the output layer parameter multiplier $\alpha_{output}$, and the attention key-projection weight multiplier $\alpha_{attn}$. See the grid and other experimental details in Appendix H.1.

We compare transferring from the 0.25x model with tuning the 1x model while controlling the total tuning budget in FLOPs.[14] To improve the reproducibility of our result: 1) we repeat the entire HP search process (a *trial*) 25 times for each setup, with number of samples as indicated in Table 4, and report the 25th, 50th, 75th, and 100th percentiles in BLEU score; 2) we evaluate each selected HP combination using 5 random initializations and report the mean performance.[15]

We pick the HP combination that achieves the lowest validation loss[16] for each trial. The reported best outcome is chosen according to the validation loss during tuning. We compare against the default in `fairseq`, which is presumably heavily tuned. The result is shown in Table 4.

---

[11]"2 layers" means the model has 2 self-attention blocks. To compare with SP Transformer, see Fig. 18.

[12]in fact, post-layernorm Transformers are much more sensitive to HPs than pre-layernorm, so our technique is more crucial for them, especially for transfer across width. Fig. 1 uses post-layernorm.

[13]`https://github.com/pytorch/fairseq/blob/master/examples/translation/README.md`.

[14]Ideally we would like to measure the wall clock time used for tuning. However, smaller models such as the proxy Transformer used for IWSLT are not efficient on GPUs, so wall clock time would not reflect the speedup for larger models like GPT-3. Thus, we measure in FLOPs, which is less dependent on hardware optimization.

[15]We do not report the standard deviation over random initializations to avoid confusion.

[16]We find this provides more reliable result than selecting for the best BLEU score.

Table 4: **Transformer on IWSLT14 De-En.** 1x and 0.25x refers to scaling of width only. Compared to traditional tuning ("Tuning on 1x"), $\mu$transfer from 0.25x provides better and more reliable outcome given fixed amount of compute. On the other hand, naive transfer (i.e. with SP instead of $\mu$P) fails completely. The percentiles are over independent trials, with each trial involving the entire tuning process with a new HP random search.

| Setup | Total Compute | #Samples | Val. BLEU Percentiles | | | |
|---|---|---|---|---|---|---|
| | | | 25 | 50 | 75 | 100 |
| `fairseq`[25] default | - | - | - | - | - | 35.40 |
| Tuning on 1x | 1x | 5 | 33.62 | 35.00 | 35.35 | 35.45 |
| Naive transfer from 0.25x | 1x | 64 | | training diverged | | |
| $\mu$Transfer from 0.25x (Ours) | 1x | 64 | **35.27** | **35.33** | **35.45** | **35.53** |

**Performance Pareto Frontier** The result above only describes a particular compute budget. Is $\mu$Transfer still preferable when we have a lot more (or less) compute? To answer this question, we produce the compute-performance Pareto frontier in Fig. 5(left), where we repeat the above experiment with different compute budgets. Evidently, our approach completely dominates conventional tuning.

**Sample Quality of Proxy Model vs Target Model** The Pareto frontier in Fig. 5(right) suggests that, given a fixed number of random *samples* from the HP space, 1) tuning the target model directly yields slightly better results than tuning the proxy model (while taking much more compute of course), but 2) this performance gap seems to vanish as more samples are taken. This can be explained by the intuition that the narrower proxy model is a "noisy estimator" of the wide target model [45].With few samples, this noise can distort the random HP search, but with more samples, this noise is suppressed.

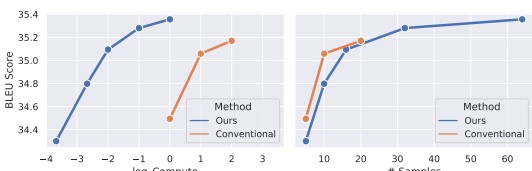

Figure 5: **Efficiency-performance Pareto frontier** of $\mu$Transfer compared to conventional tuning, on IWSLT Transformer, using random HP search as the base method. We plot the *median* BLEU score over 25 trials (Left) against relative compute budget in log scale and (Right) against number of HP samples taken. While with the same number of samples, $\mu$Transfer slightly underperforms conventional tuning, this gap vanishes with more samples, and in terms of compute, our Pareto frontier strongly and consistently dominates that of conventional tuning. Note that, in larger models (e.g. BERT or GPT-3, not shown here), we believe our efficiency advantage will only widen as our small proxy model can stay the same size while the target model grows.

## 6.2 Transformer on WMT14 En-De

We scale up to WMT14 En-De using the large (post-layernorm) Transformer from [37] with 211M parameters. We tune on a proxy model with 15M parameters by shrinking $d_{model}$, $d_{ffn}$, and $n_{head}$. For this experiment, we tune via random search the learning rate $\eta$, the output layer parameter multiplier $\alpha_{output}$, and the attention key-projection weight multiplier $\alpha_{attn}$ following the grid in Appendix H.2. The result is shown in Table 5: While random search with 3 HP samples far underperforms the `fairseq` default, we are able to match it via transfer using the same tuning budget.

Table 5: **Transformers on WMT14 En-De.** 1x and 0.25x refers to scaling of width only. We report BLEU fluctuation over 3 independent trials, i.e., 3 independent random HP searches.

| Setup | Total Compute | #Samples | Val. BLEU Percentiles | | |
|---|---|---|---|---|---|
| | | | Worst | Median | Best |
| `fairseq`[25] default | - | - | - | - | 26.40 |
| Tuning on 1x | 1x | 3 | | training diverged | 25.69 |
| Naive transfer from 0.25x | 1x | 64 | | training diverged | |
| $\mu$Transfer from 0.25x (Ours) | 1x | 64 | **25.94** | **26.34** | **26.42** |

## 6.3 BERT

Finally, we consider large-scale language model pretraining where HP tuning is known to be challenging. Using Megatron (pre-layernorm) BERT [32] as a baseline, we hope to recover the performance of the published HPs by only tuning a proxy model that has roughly 13M parameters, which we call *BERT-prototype*. While previous experiments scaled only width, here we will also scale depth, as discussed in Section 5 and validated in Fig. 4. We use a batch size of 256 for all runs and follow the standard finetuning procedures. For more details on BERT-prototype, what HPs we tune, and how we finetune the trained models, see Appendix H.3.

During HP tuning, we sample 256 combinations from the search space and train each combination on BERT-prototype for $10^5$ steps. The total tuning cost measured in FLOPs is roughly the same as training 1 BERT-large for the full $10^6$ steps; the exact calculation is shown in Appendix H.3. The results are shown in Table 6. Notice that on BERT-large, we obtain sizeable improvement over the well-tuned Megatron BERT-large baseline.

Table 6: **BERT pretraining.** HP transfer outperforms published baselines without tuning the full model directly at all. We tune BERT-base and BERT-large simultaneously via a single proxy model, *BERT-prototype*. The total tuning cost = the cost of pretraining a single BERT-large. *Model speedup* refers to the training speedup of BERT-prototype over BERT-base or BERT-large. *Total speedup* in addition includes time saving from transferring across training steps. Both speedups can be interpreted either as real-time speedup on V100s or as FLOPs speedup (which turn out to be empirically very similar in this case).

| Model | Method | Model Speedup | Total Speedup | Test loss | MNLI (m/mm) | QQP |
|-------|--------|---------------|---------------|-----------|-------------|-----|
| $\text{BERT}_{base}$ | Megatron Default | 1x | 1x | 1.995 | 84.2/84.2 | 90.6 |
| $\text{BERT}_{base}$ | Naive Transfer | 4x | 40x | | training diverged | |
| $\text{BERT}_{base}$ | $\mu$Transfer (Ours) | 4x | 40x | **1.970** | **84.3/84.8** | **90.8** |
| $\text{BERT}_{large}$ | Megatron Default | 1x | 1x | 1.731 | 86.3/86.2 | 90.9 |
| $\text{BERT}_{large}$ | Naive Transfer | 22x | 220x | | training diverged | |
| $\text{BERT}_{large}$ | $\mu$Transfer (Ours) | 22x | 220x | **1.683** | **87.0/86.5** | **91.4** |

## 6.4 GPT-3

In order to further verify $\mu$Transfer at scale, we applied it to GPT-3 6.7B [6]. This Transformer model (the *target model*) consists of 32 residual blocks with width 4096. We form the small *proxy model* by shrinking width to 256, resulting in roughly 40 million trainable parameters, 168 times smaller than the target model. HPs were then determined by a random search on the proxy model. The total tuning cost was only 7% of total pretraining cost. Details of the HP sweep can be found in Appendix H.4.

In order to exclude code difference as a possible confounder, we also re-trained GPT-3 6.7B from scratch using the original HPs from [6]. During training of the $\mu$Transfer model we encountered numerical issues that lead to frequent divergences. In order to avoid them, the model was trained using FP32 precision, even though the original 6.7B model and our re-run were trained using FP16.[17] [18] The resulting $\mu$Transfer model outperforms the 6.7B from [6], and is in fact comparable to the twice-as-large 13B model across our evaluation suite (see Table 9). Selected evaluation results can be found in Table 7 and further details are given in Table 8 and Appendix H.4.

## 7 Related Works

**Hyperparameter Tuning** Many have sought to speed up HP tuning beyond the simple grid or random search, such as via Bayesian optimization [34, 35] or multi-arm bandits [15, 18]. There are also dedicated tools such as Optuna [4] and Talos [3] which integrate with existing deep learning frameworks and provide an easy way to apply more advanced tuning techniques. Our work is

---

[17]While we are mainly focused on the efficacy of $\mu$Transfer regardless of precision, it would be interesting to ablate the effect of precision in our results, but we did not have enough resources to rerun the baseline in FP32

[18]It is quite interesting that $\mu$Transfer identified a useful region of hyperparameters leading to much improved performance, which probably would be difficult to discover normally because 1) researchers usually change hyperparameters to accomodate precision and 2) there was no precise enough justification to go against this judgment until $\mu$Transfer.

Table 7: **GPT-3 6.7B Pretraining.** Selected evaluation results for the GPT-3 6.7B model tuned with $\mu$Transfer (transfered from a small proxy model of 40M parameters), compared to the results published in [6] and a re-run with original HPs. Note that the perplexities in this table are based on a custom tokenization and are not comparable to the literature. The validation loss refers to the loss achieved on a random held-out part of our dataset. *Zero-shot*, *One-Shot* and *Few-Shot* refer to the number of additional query and answer pairs passed in the context when performing the sampling-based evaluations. See Appendix H.4 for full evaluation results.

| Task | Metric | 6.7B+$\mu$P | 6.7B re-run | 6.7B from [6] |
|------|--------|-------------|-------------|---------------|
| Validation loss | cross-entropy | **1.98** | 2.03 | - |
| PTB | perplexity | **11.4** | 13.0 | - |
| WikiText-103 | perplexity | **8.56** | 9.13 | - |
| One Billion Words | perplexity | **20.5** | 21.7 | - |
| LAMBADA Zero-Shot | accuracy | **73.5** | 70.8 | 70.3 |
| LAMBADA One-Shot | accuracy | **69.9** | 64.8 | 65.4 |
| LAMBADA Few-Shot | accuracy | 74.7 | 77.1 | **79.1** |
| HellaSwag Zero-Shot | accuracy | **72.0** | 66.7 | 67.4 |
| HellaSwag One-Shot | accuracy | **71.1** | 65.9 | 66.5 |
| HellaSwag Few-Shot | accuracy | **72.4** | 66.4 | 67.3 |

complementary to the above, as they can be used to tune the proxy model. it is only for scientific reasons that we primarily did random search throughout this work.

**Hyperparameter Transfer**  Many previous works explored transfer learning of HP tuning (e.g. [12, 28, 36, 49]). However, to the best of our knowledge, our work is the first to explore *zero-shot* HP transfer. In addition, we focus on transferring across model scale rather than between different tasks or datasets. Some algorithms like Hyperband [19] can leverage cheap estimates of HP evaluations (like using a small model to proxy a large model) but they are not zero-shot algorithms, so would still be very expensive to apply to large model training. Nevertheless, all of the above methods are complementary to ours as they can be applied to the tuning of our proxy model.

**Previously Proposed Scaling Rules of Hyperparameters**  [11, 23, 31, 33] investigated the right way to scale learning rate with batch size, with sometimes conflicting proposals. which we summarize in Appendix F. [26] studied how learning rate (and batch size) should scale with width for MLPs and CNNs trained with SGD in NTK or standard parametrizations. We provide a detailed comparison of our work with theirs in Appendix F.

Many previous works proposed different initialization or parametrizations with favorable properties, such as better stability for training deep neural networks [5, 10, 13, 21, 30, 47, 48, 51]. Our work differs from these in that we focus on the transferability of optimal HPs from small models to large models in the same parametrization.

# 8 Conclusion

Leveraging the discovery of a feature learning neural network infinite-width limit, we hypothesized and verified that the HP landscape across NNs of different width is reasonably stable if parametrized according to Maximal Update Parametrization ($\mu$P). We further empirically showed that it's possible to transfer across depth, batch size, sequence length, and training time, with a few caveats. This allowed us to indirectly tune a very large network by tuning its smaller counterparts and transferring the HPs to the full model.

**Venues of Improvement**  Nevertheless, our method has plenty of room to improve. For example, initialization does not transfer well across depth, and depth transfer generally still does not work for post-layernorm Transformers. This begs the question whether a more principled parametrization in depth could solve these problems. Additionally, Fig. 4 shows that the optimal HP still shifts slightly for smaller models. Perhaps by considering finite-width corrections to $\mu$P one can fix this shift. Finally, it will be interesting to consider if there's a way to transfer regularization HPs as a function of both the model size and data size.

**Broader Impact**  Our work makes HP tuning of large models more efficient. This enables large models to be better tuned given the same compute budget, thereby increasing the performance per cost. Organizations large and small can focus their research on small models and scale up only once with reasonable confidence that the training would go well. We do not foresee any direct negative societal impact.

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
