# A   Practical Considerations

In this section, we outline several useful tips and tricks that can improve the quality of hyperparameter transfer in practice.

## A.1   Zero Initialization for Output Layers

We find that the optimal hyperparameters of small and large width models match more closely when we initialize output layers at 0 (i.e. with variance $\sigma^2/\text{fan\_in}$ where $\sigma = 0$ instead of positive $\sigma$). This is because the neural network in $\mu$P is approximately a Gaussian process (GP) at initialization with variance on the order $\Theta(\sigma^2/width)$ (contrast this with SP networks, which approximates a GP with $\Theta(\sigma^2)$ variance) [41, 45]. Of course, when width is large, this variance vanishes, but this can be far from so in the small proxy model. This discrepancy in the initial GP can cause the training trajectory of the proxy model to be very different from the trajectory of the large target model, causing a mismatch in the optimal hyperparameters. By initializing the output layer at 0, we remove this mismatch in the initial GP. Empirically we do not find this modification to be detrimental to performance.

## A.2   Activation Functions

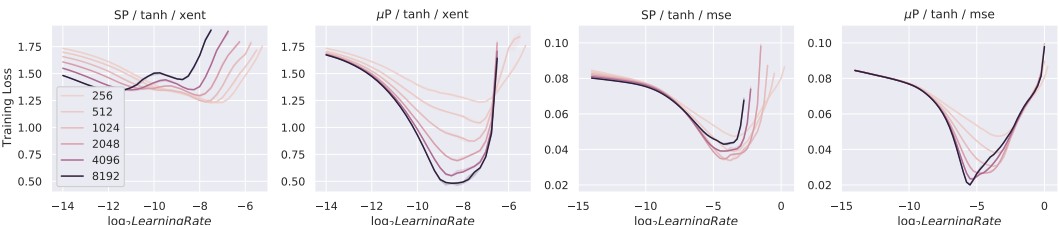

Figure 6: **Squashing activation functions reduce transfer quality.** MLP of different hidden sizes with `tanh` activation trained for 20 epoch on CIFAR-10 using SGD. Left uses cross-entropy as loss function; right uses mean square error; columns alternate between standard parametrization (SP) and maximal update parametrization ($\mu$P). Compared to ReLU, `tanh` exhibits slower convergence for $\mu$P, yet it still outperforms SP when width is increased

When the network is narrow, its approximation to the infinite-width behavior becomes crude, which is manifested as large fluctuations in preactivation coordinates. When using a squashing activation functions like `softmax` or `tanh`, this causes narrower networks to saturate the activation more than wider ones, which results in a systematic bias in the gradients and therefore the hyperparameter landscape. This can be seen in Fig. 6, where we use `tanh` as the network activation function.

Therefore, we recommend replacing non-essential squashing activation functions with `ReLU`, whose derivative depends only on the sign of the pre-activation. A similar reasoning can be applied to superlinear activation functions, where the distribution of activation values can have heavy tails, leading to slow convergence to the infinite-width limit. However, such activations are rarely used in practice.

## A.3   Enlarge $d_k$

We find that small $d_{head} = d_k$ can lead to a highly noisy HP landscape, as shown in Fig. 7. This can significantly decrease the quality of random HP search on the small proxy model. To solve this, we find it useful to decouple $d_k$ from $d_{model}$ (so that $d_{model} \neq d_k \cdot n_{head}$) and maintain a relatively large $d_k$ even as $d_{model}$ is shrunk in the proxy model. For example, pegging $d_k = 32$ is generally effective. Training or inference speed are not usually affected much by the larger $d_k$ because of CUDA optimizations. By Appendix G.2, this decoupling of $d_k$ from $d_{model}$ is theoretically justified, and as shown in Fig. 7, it significantly denoises the HP landscape.

## A.4   Non-Gaussian vs Gaussian Initialization

We find non-Gaussian (e.g. uniform) initialization can sometimes cause wider models to perform worse than narrower models, whereas we do not find this behavior for Gaussian initialization. This is

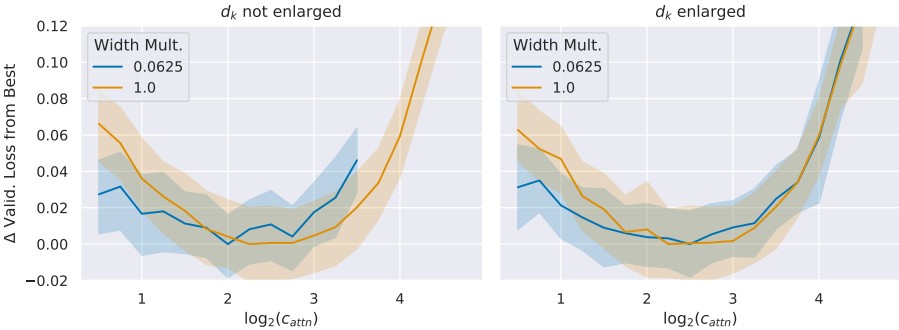

Figure 7: **Enlarging $d_k$ makes $\mu$Transfer more precise.** Here we plot all curves *after subtracting their minima* for easier visual comparison. Transformer on IWSLT 14 similar to the setup in Appendix H.1 where the $d_{model} = 512$ for a width multiplier of 1, $n_{head} = 4$, and $d_q = d_k$. **(Left)** We leave $d_q = d_k = d_{model}/n_{head}$, so $d_k = 8$ for width-multiplier 0.0625. The optimum for the attention logit multiplier $c_{attn}$ is noisy and does not accurately transfer across width. **(Right)** We enlarge $d_q = d_k$ to a minimum of 128. The HP landscape is much smoother than in (Left), and the optima align between narrow and wide models.

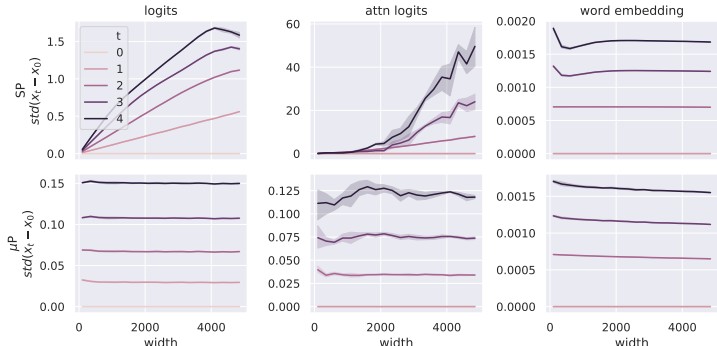

Figure 8: **Logits and attention logits, but not word embeddings, of a Transformer blow up with width in SP after 1 step of training.** In contrast, all three are well-behaved with width in $\mu$P. Here we measure how much different values change coordinatewise from initialization over 4 steps of Adam updates, as a function of width. Specifically, we plot the standard deviation of the coordinates of $x_t - x_0$, for $t = 0, \ldots, 4$, and $x \in \{$logits, attention logits, word embeddings$\}$, where $t = 0$ indicates initialization.

consistent with theory, since in the large width limit, one should expect non-Gaussian initialization will behave like Gaussian initializations anyway (essentially due to Central Limit Theorem, or more precisely, universality), but the non-Gaussianity slows down the convergence to this limit.

### A.5  Using a Larger Sequence Length

For Transformers, we empirically find that we can better transfer initialization standard deviation from a narrower model (to a wide model) if we use a larger sequence length. It is not clear why this is the case. We leave an explanation to future work.

## B  The Defects of SP and How $\mu$P Fixes Them

The question of SP vs $\mu$P has already been studied at length in [45]. Here we aim to recapitulate the key insights, with more explanations given in Appendix L.3.

**An Instructive Example**   As shown in [45] and Appendix L.3, in SP, the network output will blow up with width after 1 step of SGD. It's instructive to consider a 1-hidden-layer linear perceptron $f(x) = V^\top U x$ with scalar inputs and outputs, as well as weights $V, U \in \mathbb{R}^{n \times 1}$. In SP, $V_\alpha \sim$

$\mathcal{N}(0, 1/n)$ ad $U_\alpha \sim \mathcal{N}(0, 1)$ for each $\alpha \in [n]$. This sampling ensures that $f(x) = \Theta(|x|)$ at initialization. After 1 step of SGD with learning rate 1, the new weights are $V' \leftarrow V + \theta U, U' \leftarrow U + \theta V$, where $\theta$ is some scalar of size $\Theta(1)$ depending on the inputs, labels, and loss function. But now

$$f(x) = V'^\top U'x = (V^\top U + \theta U^\top U + \theta V^\top V + \theta^2 U^\top V)x \tag{5}$$

blows up with width $n$ because $U^\top U = \Theta(n)$ by Law of Large Numbers.

Now consider the same network in $\mu$P. According to Table 3, we now have $V_\alpha \sim \mathcal{N}(0, 1/n^2)$ in contrast to SP, but $U_\alpha \sim \mathcal{N}(0, 1)$ as before, with learning rates $\eta_V = 1/n, \eta_U = n$. After 1 step of SGD, we now have

$$f(x) = (V^\top U + \theta n^{-1} U^\top U + \theta n V^\top V + \theta^2 U^\top V)x,$$

and one can verify this is $\Theta(1)$ and thus does not blow up with width.[19]

**Some Layers Update Too Fast, Others Too Slow**  One can observe the same behavior in more advanced architectures like Transformers and optimizers like Adam; in fact, in SP, other hidden quantities like attention logits will also blow up with width after 1 step, but in $\mu$P still remain bounded, as shown in Fig. 8(middle).

One might think scaling down the learning rate with width can solve this problem in SP. However, other hidden activations like the word embedding (Fig. 8(right)) in a Transformer update by a width-independent amount for each step of training, so scaling down the learning rate will effectively mean the word embeddings are not learned in large width models. Similar conclusions apply to other models like ResNet (in fact, one can observe in the SP linear MLP example above, the input layer is updated much more slowly than the output layer). On the other hand, $\mu$P is designed so that all hidden activations update with the same speed in terms of width (see Appendix L.2 for why).

**Performance Advantage of $\mu$P**  This is why a wide model tuned with $\mu$Transfer should in general outperform its SP counterpart with (global) learning rate tuned. For example, this is the case for the width-8192 Transformer in Fig. 1, where, in SP, the optimal learning rate needs to mollify the blow-up in quantities like logits and attention logits, but this implies others like word embeddings do not learn appreciably. This performance advantage means $\mu$Transfer does more than just predicting the optimal learning rate of wide SP models. Relatedly, we observe, for any fixed HP combination, training performance never decreases with width in $\mu$P, in contrast to SP (e.g., the $\mu$P curves in Figs. 1, 3 and 16 do not cross, but the SP curves do; see also Appendix D).

## C  Parametrization Matters: A Primer for Multiple Hyperparameters

Here we give more intuition why we need to reparametrize *all* hyperparameters. In practice, neural networks have multitudes of hyperparameters all interacting together. In our example of Section 2, hyperparameter optimization would be akin to minimizing the function[20]

$$F_n(c^1, \ldots, c^k) \stackrel{\text{def}}{=} \mathbb{E}_{x_1, \ldots, x_n} f((c^1 + \cdots + c^k)(x_1 + \cdots + x_n)).$$

where $x_1, \ldots, x_n$ are as in Eq. (1) and $c^1, \ldots, c^k$ are analogous to $k$ hyperparameters. For the same reasoning in Section 2, the *correct parametrization* is in $(\alpha^1, \ldots, \alpha^k)$ where $\alpha^i = c^i \sqrt{n}$.

While this is straightforward, in practice, researchers often fix some hyperparameters (e.g., they tune only learning rate but neglects to scale parameter multipliers or initialization correctly). For example, if we only partially reparametrize and optimize in $\alpha^1$ while fixing $c^2, \ldots, c^k$, then the optimal $\alpha^1$ is $(\alpha^1)^* = \alpha^* - (c^1 + \ldots + c^k)\sqrt{n}$ where $\alpha^*$ is the optimal $\alpha$ for Eq. (1). Thus, as $n \to \infty$, $(\alpha^1)^*$ still blows up even though we parametrized $\alpha^1$ correctly. More generally, the incorrect parametrization of some hyperparameters forces other hyperparameters to increasingly compensate for it as width grows, distorting their optima, even if the latter are correctly parametrized.

---

[19]Note in this example, Glorot initialization [10] (i.e. with variance $1/(\text{fan\_in} + \text{fan\_out})$) would scale asymptotically the same as $\mu$P and thus is similarly well-behaved. However, if one adds layernorm or batchnorm, then Glorot will cause logit blowup like SP, but $\mu$P still will not.

[20]Here, for simplicity of the example, we model the interaction between "hyperparameters" $c^1, \ldots, c^k$ as additive, but in real neural networks such interactions are usually much more complicated.

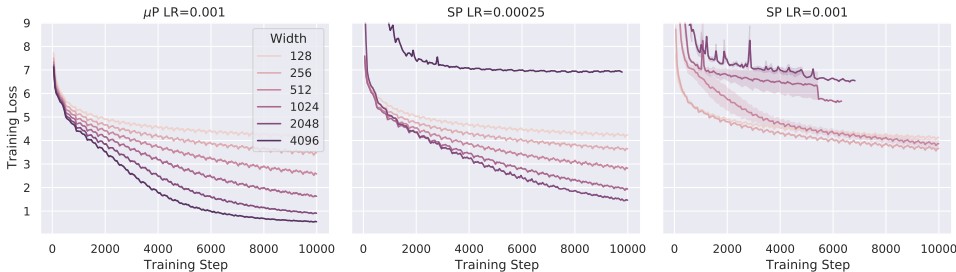

Figure 9: **Wider is always better in training loss under $\mu$P, but not in SP, given the same HP.**
Learning curves for $\mu$P and SP with different learning rates, aggregated over 5 seeds. **(Left)** Wider
$\mu$P models always achieve better training loss at any time in training. **(Middle)** If using a small
learning rate, SP models can appear to do so up to some large width, at which point the pattern fails
(at width 2048 in our plot). **(Right)** If using a large learning rate, SP model can do *worse* with width;
here the SP model is identical to the $\mu$P model in (Left) at width 128.

## D   Wider is Better in $\mu$P *Throughout Training*

In earlier plots like Figs. 1 and 3, we saw that
at the end of training, wider is always better
in $\mu$P but not in SP. In fact, we find this to
be true *throughout training*, as seen in Fig. 9,
modulo noise from random initialization and/or
data ordering, and assuming the output layer is
zero-initialized (which has no impact on perfor-
mance as discussed in Appendix A.1). We then
stress-tested this on a $\mu$P GPT-3 Transformer
(on the GPT-3 training data) by scaling width
from 256 to 32,768 using a fixed set of HPs
(Fig. 10). Wider models consistently match
or outperform narrower models at each point
in training (except a brief period around 1e8
training tokens, likely due to noise because
we ran only 1 seed due to computational cost).
Our observation suggests that wider models are
strictly more data-efficient if scaled appropri-
ately. By checking "wider-is-better" early in
training, one can also cheaply debug a $\mu$P implementation.

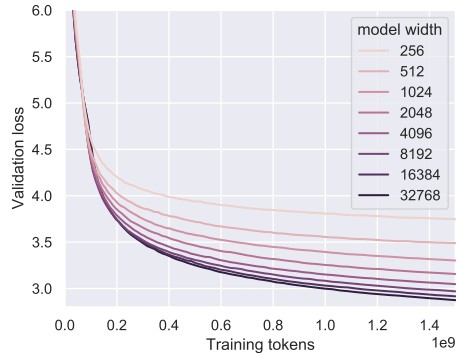

Figure 10: **Stress-testing "wider-is-better" in $\mu$P.**
Here we trained a GPT-3 transformer with 4 layers
and widths from 256 to 32,768. Modulo a brief
period around 1e8 training tokens, wider is better
throughout training.

## E   *Useful* Hyperparameter Transfer: A Theoretical Puzzle

We want to tune HPs on a small model with width $N$ such that its HP landscape looks like that of
a large model with width $\gg N$. Our intuition in Section 2 and Appendices C and L leads us to $\mu$P.
However, for this to be useful, we *do not want* the small model (as a function) after training to be
close to that of the large model — otherwise there is no point in training the large model to begin
with. So $N$ 1) must be large enough so that the HP optimum converges, but 2) cannot be so large
that the functional dynamics (and the loss) converges. The fact that such $N$ exists, as demonstrated
by our experiments, shows that: In some sense, the HP optimum is a "macroscopic" or "coarse"
variable which converges quickly with width, while the neural network function (and its loss) is a very
"microscopic" or "fine" detail that converges much more slowly with width. However, theoretically,
it is unclear why this should happen, and where else we should expect such *useful* HP transfer. We
leave an explanation to future work.

# F Detailed Discussions on Related Works

## F.1 Hyperparameter Tuning

Many have sought to speedup HP tuning beyond the simple grid or random search. Snoek et al. [34] treated HP tuning as an optimization process and used Bayesian optimization by treating the performance of each HP combination as a sample from a Gaussian process (GP). Snoek et al. [35] further improved the runtime by swapping the GP with a neural network. Another thread of work investigated how massively parallel infrasture can be used for efficient tuning under the multi-arm bandit problem [15, 18]. There are also dedicated tools such as Optuna [4] and Talos [3] which integrate with existing deep learning frameworks and provide an easy way to apply more advanced tuning techniques.

Our approach is distinct from all of the above in that it does not work on the HP optimization process itself. Instead, it decouples the size of the target model from the tuning cost, which was not feasible prior to this work. This means that **no matter how large the target model is, we can always use a fixed-sized proxy model to probe its HP landscape** Nevertheless, our method is complementary, as the above approaches can naturally be applied to the tuning of the proxy model; it is only for scientific reasons that we use either grid search or random search throughout this work.

## F.2 Previously Proposed Scaling Rules of Hyperparameters

**(Learning Rate, Batch Size) Scaling**   [33] proposed to scale learning rate with batch size while fixing the total epochs of training; [11] proposed to scale learning rate as $\sqrt{batchsize}$ while fixing the total number of steps of training. However, [31] showed that there's no consistent (learning rate, batch size) scaling law across a range of dataset and models. Later, [23] studied the trade-off of training steps vs computation as a result of changing batch size. They proposed an equation of $a/(1 + b/batchsize)$, where $a$ and $b$ are task- and model-specific constants, for the optimal learning rate (see their fig 3 and fig 5). This law suggests that for sufficiently large batch size, the optimal learning rate is roughly constant.[21] This supports our results here as well as the empirical results in [31, fig 8].

**Learning Rate Scaling with Width**   Assuming that the optimal learning rate should scale with batch size following [33], [26] empirically investigated how the "noise ratio" $LR/batchsize$ scales with width for MLP and CNNs in NTK parametrization (NTP) or standard parametrization (NTP) trained with SGD. They claimed that, in networks without batch normalization, the optimal noise ratio is constant in SP but scales like $1/width$ for NTP. However, they found this law breaks down for networks with normalization.

Here in our work, Fig. 3 contradicts their results on SP MLP by showing the optimal learning rate (fixing batch size) shifts with width. We believe this difference is 1) due to their erroneous assumption that optimal learning rate scales with batch size (as debunked by [23, 31]) and 2) because their SP experiments were done by fixing the learning rate and only sweeping batch size.

Furthermore, Fig. 1 clearly shows the optimal learning rate is *not* constant in SP for Transformers (trained with Adam). Other differences in our works include our applicability to 1) networks with normalization, 2) Adam and other adaptive optimizers, 3) our empirical validation of transfer across depth and sequence length, and 4) explicit validation of tuning via $\mu$Transfer on large models like BERT-large.

Finally, as argued in [45] and Appendix L.3, SP and NTP lead to bad infinite-width limits in contrast to $\mu$P and hence are suboptimal for wide neural networks. For example, sufficiently wide neural networks in SP and NTP would lose the ability to learn features, as concretely demonstrated on word2vec in [45].

**Input Layer Parametrization**   While typically, the input layer is initialized with fanin initialization, in language models where the input and output layers are shared (corresponding to word embeddings), it can actually be more natural to use a fanout initialization (corresponding to fanin initialization of

---

[21]while the optimal learning is roughly linear in batch size when the latter is small

the output layer). In fact, we found that `fairseq` [25] by default actually implements our proposed input layer parametrization (both the fanout initialization and the $\sqrt{\text{fan\_out}}$ multiplier).[22]

**From the Theory of Infinite-Width to the Practice of Finite-Width Neural Networks and Back**
[45] introduced $\mu$P as the unique parametrization that enables all layers of a neural network to learn features in the infinite-width limit, especially in contrast to the NTK parametrization [14] (which gives rise to the NTK limit) that does not learn features in the limit. Based on this theoretical insight, in Appendix L.3, we argue that $\mu$P should also be the *unique* parametrization that allows HP transfer across width; in short this is because it both 1) preserves feature learning, so that performance on feature learning tasks (such as language model pretraining) does not become trivial in the limit, and 2) ensures each parameter tensor is not stuck at initialization in the large width limit, so that its learning rate does not become meaningless. At the same time, our results here suggest that $\mu$P is indeed the *correct* parametrization for large neural networks and thus provide empirical motivation for the theoretical study of the infinite-width $\mu$P limit.

# G   Which Hyperparameters Can Be Transferred? (Continued)

## G.1   Further Discussions on Hyperparameter Categories

Below, we discuss the reasoning behind each kind, which are supported by our empirical evidence collected in Fig. 4 on Transformers as well as those in Appendix I.1 on ResNet.

**Transferable Hyperparameters**   In Table 2, we summarize which HPs can be transferred across training scale. The transfer across *width*, as explained in Section 2, is theoretically justified, while we present the transfer across the other dimensions as empirical results.

These cover most of the well-known and important HPs when the need for regularization is not paramount, e.g., during large scale language model pretraining. Parameter Multipliers are not well-known HPs, yet we include them here as they serve a bridge between SP and $\mu$P and can impact model performance in practice. Concretely, any SP and $\mu$P neural networks of the same width can have their Parameter Multipliers tuned so that their training dynamics become identical.

**Hyperparameters That Don't Transfer Well**   Not all HPs transfer well even if we use $\mu$P. In particular, those whose primary function is to regularize training to mitigate "overfitting" tend not to transfer well. Indeed, intuitively, regularization needs to be applied more heavily in larger models, so naturally we do not expect the same regularization HPs to stay constant across model sizes.

To the best of our knowledge, there is no strict separation between HPs that regularize and those that don't. However, conventional wisdom tells us that there exists a spectrum of how much regularizing effect a HP has. For example, dropout probability and weight decay are among those whose primary function is to regularize, whereas batch size and learning rate might regularize training in some cases but affect the dynamics more so in other ways. Our empirical exploration tells us that the former do not transfer well, while the latter do. Our subsequent discussion will focus on the latter; we leave to future works the expansion to the former.

**Hyperparameters Transfered *Across***   We have left out a category of HPs that defines the training *scale*, or in practical terms, training cost. This includes 1) those that define how many operations a model's forward/backward pass takes, such as the model's width, depth, and in the case of language modeling, sequence length; and 2) those that define how many such passes are performed, such as batch size and number of training steps.

As recent works have shown [6, 16], improvements along any of these *scale* dimensions lead to apparently sustainable gain in performance; as a result, we are primarily interested in transferring other HPs *across* these dimensions that define scale, rather than finding the optimal scale.[23] This category of HPs is particularly crucial as one can speedup training by downsizing in one or multiple

---

[22]But it certainly does not implement other parts of our parametrization, like Adam learning rate scaling or the output multiplier.

[23]In particular, we are not fixing the total training FLOPs when we scale, which requires understanding the tradeoff of different scale HPs. For example, when we transfer across batch size, we *fix* the number of steps of training (*not* the number of epochs), so that the total FLOPs scales linearly.

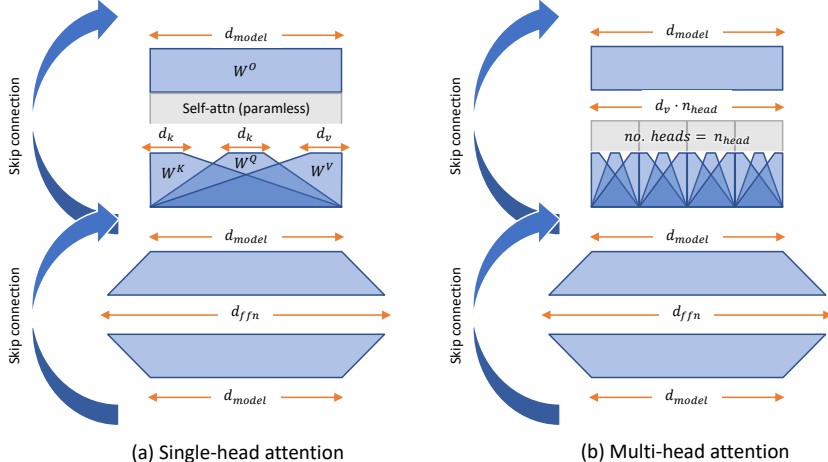

(a) Single-head attention         (b) Multi-head attention

Figure 11: Schematics of each Transformer layer. Commonly, the key and value dimensions $d_k$ and $d_v$ are both set to $d_{model}/n_{head}$, and this is referred to as $d_{head}$.

such dimensions. Indeed, it's very common for practitioners to implicitly transfer HPs across the number of training samples by tuning on only a subset of the full training data.

Our insights from the infinite-width limit inspired us to explore HP tranfer across *width*, which does not work under SP as we have shown earlier. Building upon our success with width, which is well explained theoretically, we hope to push the limit of compute-saving by investigating the other dimensions empirically. To the best of our knowledge, the transferability of optimal HPs across depth, batch size, sequence length, and training time has not been rigorously investigated previously, with the main exception of the literature on (learning rate, batch size) scaling [31, 33] where our transferability result of learning rate across batch size recapitulates [23].[24] See Appendix F.2 on how our results relate to prior works. We will primarily focus on the Transformer architecture in the main text with evidence for ResNet in Appendix I.1.

### G.2    On the Definitions of Width

Our theory allows more general notions of width. This is especially relevant in Transformers, where $d_{model}, d_{head} = d_k, d_v, n_{head}, d_{ffn}$ (see Fig. 11) can all be construed as measures of width. We briefly discuss these here, with more theoretical justification in Appendix L.2.1 and empirical validation below.

**Varying Width Ratio**    So far we have assumed that every hidden layer is widened by the same factor. But in fact we can widen different hidden layers differently. This is useful, for example, in a Transformer where we may want to use a smaller $d_{ffn}$ during tuning. If we are using Adam, as long as the width of every layer still tends to infinity, we still obtain approximately the same limit[25], so the $\mu$Transfer remains theoretically justified.

See Fig. 12 for an empirical validation on IWSLT-14 using a Transformer.

**Number of Attention Heads**    In attention-based models, one typically splits hidden size into multiple attention heads following $d_{model} = d_{head} \times n_{head}$. So far we have assumed $d_{head}$ and $d_{model}$ to be width, but it's possible and potentially advantageous to fix $d_{head}$ and treat $n_{head}$ as the width, or increasing both simultaneously. This allows our technique to handle many popular models, including GPT-3 [6], which scale up by fixing $d_{head}$ and increasing $n_{head}$. See Fig. 13 for an empirical validation on Wikitext-2.

---

[24]There's also a literature on the proper initialization for training deep networks effectively (e.g. [5, 13, 21, 30, 47, 48, 51]), but they do not study the *transferability* per se. See Appendix F.2

[25]This also applies for SGD, but we need more involved scaling to keep the limit approximately the same.

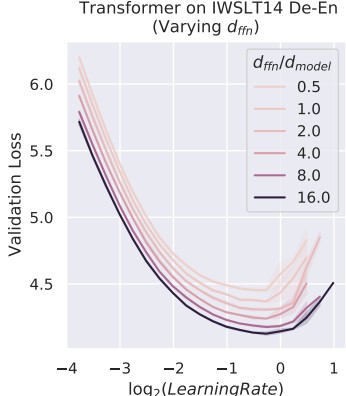

Figure 12: Learning rate landscape in $\mu$P is stable even if we vary $d_{ffn}$ by a factor of 32, fixing $d_{model}$.

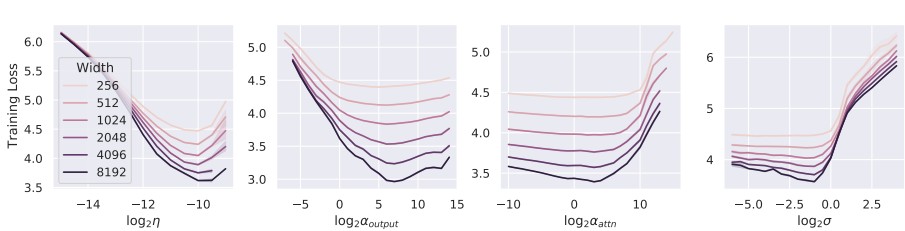

Figure 13: $\mu$Transfer across width when we fix $d_{head}$ and vary $d_{model}$ and $n_{head}$. $\alpha_{output}, \alpha_{attn}$ are multipliers for output and key weights, and $\sigma$ is initialization standard deviation.

**Varying Just the Width of Attention Heads**   A specific useful instance of varying width ratio is decoupling the key and value dimensions $d_k$ and $d_v$ and scaling $d_k$ differently from (typically larger than) $d_{model}/n_{head}$. This works as long as we use $1/d$ scaled-attention as in Definition 4.1 (instead of $1/\sqrt{d}$ as is done commonly). When tuning on the small proxy model, if $d_k$ is too small, the HP landscape can be quite noisy. Keeping $d_k$ relatively large while shrinking all other dimensions solves this problem, while still obtaining significant speedup.

# H   Experimental Details

## H.1   IWSLT

IWSLT14 De-En is a well-known machine translation benchmark. We use a Transformer implemented in `fairseq` [25] with a default $d_{model} = {}^{1}/{4}d_{ffn} = 512$ and $d_k = d_q = d_v = {}^{d_{model}}/{n_{head}} = 128$ (amounting to 40M parameters), which we denote as the *1x model*. For transfer, we tune on a proxy model with the same $n_{head}$ but with $d_{model}$ and other dimensions 4 times smaller; we will call this the *0.25x model* (but it has 4M parameters). All models are trained with Adam for 100 epochs and validated at the end of every epoch. We tune via random search the learning rate $\eta$, the output layer parameter multiplier $\alpha_{output}$, and the attention key-projection weight multiplier $\alpha_{attn}$ following the grid

- $\eta$: $5 \times 10^{-4} \times 2^z$, where $z \in \{-1.5, -1.25, -1, ..., 1.25\}$
- $\alpha_{output}$: $2^z$, where $z \in \{-8, -7, -6, ..., 7\}$
- $\alpha_{attn}$: $2^z$, where $z \in \{-3, -2, -1, ..., 8\}$

## H.2   WMT

We scale up to WMT14 En-De using the large Transformer from [37], with a $d_{model} = {}^{1}/{4}d_{ffn} = 1024$ and $d_q = d_k = d_v = {}^{d_{model}}/{n_{head}} = 64$. We use the exact same setup and reproduce their result as our baseline. Then, we build the proxy model by shrinking the target model's $d_{model}$ from

the original $1024$ to $256$, $d_{ffn}$ from $4096$ to $256$ and $n_{head}$ from $16$ to $4$. This reduces the total parameter count from $211$M to $15$M. We then perform the HP search on the proxy model and take the best according to validation loss, before testing on the target model. We tune via random search the learning rate $\eta$, the output layer parameter multiplier $\alpha_{output}$, and the attention key-projection weight multiplier $\alpha_{attn}$ following the grid

- $\eta$: $6 \times 10^{-4} \times 2^z$, where $z \in \{-1.5, -1.25, -1, ..., 1.25\}$
- $\alpha_{output}$: $2^z$, where $z \in \{-8, -7, -6, ..., 7\}$
- $\alpha_{attn}$: $2^z$, where $z \in \{-3, -2, -1, ..., 8\}$

## H.3  BERT

**Details of BERT Prototype**   Our proxy model has 10 Transformer layers with $d_{model} = d_{ffn} = 256$. We also reduce the number of attention heads to 8 with a $d_{head}$ of 32. We call it BERT Prototype since we can increase its width and depth according to our definitions to recover both BERT Base and BERT Large, which enables us to sweep HPs once and use for both models. Overall, BERT Prototype has 13M trainable parameters, a fraction of the 110M in BERT Base and the 350M in BERT Large.

**Hyperparameters Tuned for Pretraining**   We tune the following HPs for pretraining: Adam learning rate $\eta$, embedding learning rate $\eta_{emb}$, output weight multiplier $\alpha_{output}$, attention logits multiplier $\alpha_{attn}$, layernorm gain multiplier $\alpha_{LN_{gain}}$, and bias multiplier $\alpha_{bias}$.

We sample 256 combinations from the follow grid:

- $\eta$: $1 \times 10^{-4} \times 2^z$, where $z \in \{1.5, 2, 2.5, 3, 3.5\}$
- $\eta_{emb}$: $1 \times 10^{-4} \times 2^z$, where $z \in \{-1, -0.5, 0, 0.5, 1\}$
- $\alpha_{output}$: $2^z$, where $z \in \{2, 4, 6\}$
- $\alpha_{attn}$: $2^z$, where $z \in \{3, 3.5, 4, ..., 7\}$
- $\alpha_{LN_{gain}}$: $2^z$, where $z \in \{8.5, 9, 9.5, 10, 10.5\}$
- $\alpha_{bias}$: $2^z$, where $z \in \{8.5, 9, 9.5, 10, 10.5\}$

The ranges are chosen to include the implicit choices of these HPs in SP BERT Large.

**Finetuning Procedure and Hyperparameters**   We hand-pick the finetuning HPs after training the full-sized model. As regularization is an essential ingredient in successful finetuning, we do not transfer such HPs (at least via the suite of techniques presented in this work) (see Table 1). We focus on MNLI [40] and QQP, which are two representative tasks from GLUE [38]. Following [22], we used Adam [17] with a learning rate of $5 \times 10^{-5}$ and a batch size of 64. The maximum number of epochs was set to 5. A linear learning rate decay schedule with warm-up of $0.1$ was used. All the texts were tokenized using wordpieces and were chopped to spans no longer than 128 tokens.

## H.4  GPT-3

**Baseline 6.7B GPT-3 Transformer**   As the GPT-3 codebase has evolved since the publication of [6], we re-trained the 6.7B model from scratch to remove changes in our codebase as a possible confounder. The main difference to [6] is a modified learning rate decay schedule, where the learning rate is decayed to zero at the end of training rather than being decayed to 0.1 of the initial value.

**Random Search using Reduced-Width Proxy Model**   In order to find a good set of hyperparameters for the $\mu$Transfer version of the 6.7B model, we performed a hyperparameter search over a reduced version of the model (i.e., the proxy model), where the width is set to 256 hidden units. This proxy model inherits changes from the evolved GPT-3 codebase: it uses relative [8] (instead of absolute) position encoding. Early on, we noted that on the proxy model, linear learning rate decay outperformed the default cosine schedule, so all subsequent experiments for the proxy models use a linear decay schedule. By Fig. 4, $\mu$Transferring this linear decay schedule to the full model sould maintain such a performance advantage over the cosine schedule.

The hyperparameter search space consists of the following hyperparameters:

- **learning rate:** Sampled from $10^{\text{Uniform}(-4, -1)}$

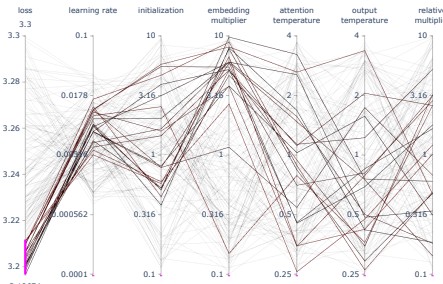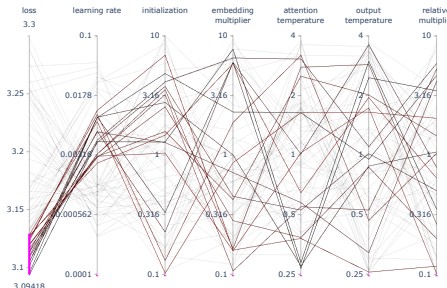

Figure 14: Results of the random search over reduced-width proxy models trained on 4 (left) and 16 (right) billion tokens. Only the best performing runs are highlighted.

- **initialization scale:** All the parameters are multiplied - sampled from $10^{\text{Uniform}(-1,1)}$
- **attention temperature:** Reciprocal of the multiplier applied to the input to attention softmax. Sampled from $4^{\text{Uniform}(-1,1)}$.
- **output temperature:** Reciprocal of the multiplier applied to the input to softmax that produces the distribution over output tokens. Sampled from $4^{\text{Uniform}(-1,1)}$.
- **embedding multiplier:** Scalar by which we multiply the output of the embedding layer. Sampled from $10^{\text{Uniform}(-1,1)}$.
- **relative position embedding multiplier:** Scalar by which we multiply vectors representing relative position. Sampled from $10^{\text{Uniform}(-1,1)}$.

In order to make the search more efficient we reduced the total number of training tokens. We hypothesized that tuning hyperparameters on a reduced total number of tokens does not significantly affect optimal hyperparameters. To verify, we trained two different horizons and compared the results. While the target model was to be trained on 300 billion tokens, we tuned the proxy model on only subsets consisting of 4 billion and 16 billion tokens. This impacts both the total training time and and the length of the linear learning rate decay schedule. Other than hyperparameters explicitly listed above and the training horizon, the rest was the same as what we intended to use for the full width 6.7B training run.

**Analyzing the Results of the Random Search**   We performed 467 training runs of the proxy model, out of which 350 were for 4 billion tokens (286 completed without diverging) and 117 for 16b tokens (80 completed without diverging). See Fig. 14 for summary of the results.

As suspected, we observed that the results are well-aligned for both 4 and 16 billion tokens versions. We observe learning rate and initialization scale impact the results the most. Based on the results we chose 0.006 for the former and 2.5 for the latter. Since most other hyperparameters appear to have negligible effect on performance, they were kept at their default values of 1, the only exception being the embedding scale, where higher values seem to perform better and it was therefore set to 10.

**Training the $\mu$Transfer Model**   We encountered frequent divergences in our initial attempt to train the $\mu$Transfer model. We traced the issue back to underflow of FP16 tensors in the backwards pass and therefore switched to training the model in FP32. This allowed us to finish the training run without divergences. We hypothesize that the divergence issue is related to $\mu$Transfer picking more aggressive hyperparameters, for example a higher learning rate on linear weight tensors compared to the original model. In order to exclude code differences as a possible confounder, we re-trained GPT-3 6.7B from scratch using the original hyperparameters. The only difference compared to the version published in [6] is that the learning rate was decayed fully, whereas the learning rate of the model from [6] was only decayed to 10% of its starting value. The retrained model performs slightly worse than the original published in [6]. We suspect that this is because it made less progress during the last phase of training where the learning rate is close to zero. The training curves of the $\mu$Transfer model and the re-run of the original 6.7B can be seen in Fig. 15. Detailed evaluation results can be found in Table 8 and Table 9.

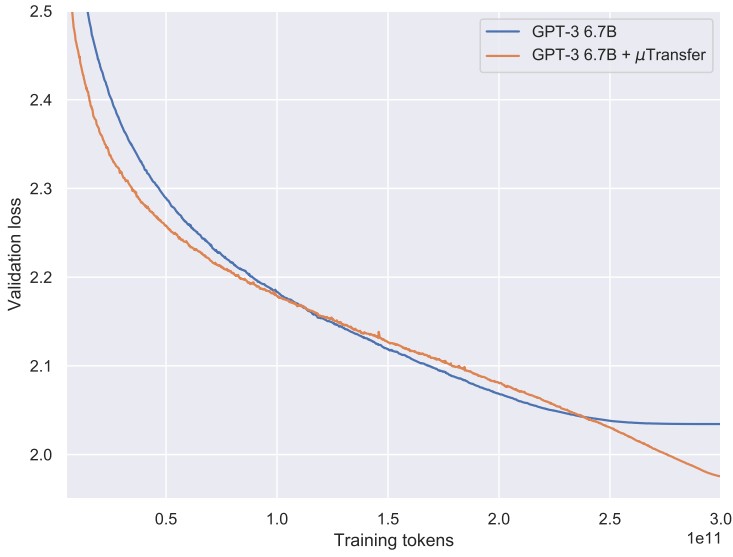

Figure 15: **The training curves of the GPT-3 6.7B model with $\mu$Transfer (orange) and a re-run with the original settings from [6] (blue).** The $\mu$Transfer model was trained using FP32 activations and weights after initially encountering stability issues with the hyperparameters computed using $\mu$P, while the re-run used the original FP16 training. The $\mu$Transfer model seems to underperform in the middle of training, but achieves a much better final validation loss once the learning rate is fully decayed. The $\mu$Transfer model uses a linear learning rate decay schedule while the original model uses a cosine schedule.

**Ratio of Tuning Cost to Pretraining Cost**    in FLOPs can be approximated as

$$\frac{s(t_1 N_1 + t_2 N_2)}{ST} \approx 0.07$$

where

- $s = 40$ Million is number of parameters of the proxy model
- $S = 6.7$ Billion is number of parameters of the target model
- $t_1 = 4$ Billion is the number of training tokens for the short horizon HP search, and $N_1 = 350$ is the corresponding number of random HP search trials.
- $t_2 = 16$ Billion is the number of training tokens for the longer horizon HP search, and $N_1 = 117$ is the corresponding number of random HP search trials.
- $T = 300$ Billion is the number of training tokens for the 6.7B target model.

Here we are using the fact that the training FLOPs of a Transformer per token is roughly proportional to its number of parameters.

## I    Additional Experiments

### I.1    Experiments on ResNets

#### I.1.1    ResNet on CIFAR-10

**Setup**    For this case we use Davidnet [2], a ResNet variant that trains quickly on CIFAR-10, so as to efficiently investigate its HP landscape. We train with SGD on CIFAR-10 for 10 epochs; all results are averaged over 15 random seeds. We use a width multiplier to identify models of different width, and a multiplier of 1 corresponds to the original model in [2]. We look at validation accuracy here as the model barely overfits, and our observations will hold for the training accuracy as well. We first

Table 8: **Full evaluation results of our GPT-3 6.7B models**: The new model tuned with $\mu$Transfer (marked $\mu P$), the original model from [6], and a re-training of this model from scratch with the original hyperparameter settings (marked *re-run*). The sampling-based evaluations shown here are a subset of the ones from [6]. Since the sampling-based evaluations are subject to high variance, Wikitext 103 and the LM1B benchmark have been added to help distinguish the relative performance of the $\mu P$ and non-$\mu P$ model. Note that Wikitext-103 [24] and the LM1B [7] benchmarks overlap with the training dataset. Accuracies and F1 scores have been multiplied by 100. The perplexities reported in this table are based on a custom BPE encoding and are not comparable to other results in the literature. The number $k$ of examples in the context for each task is identical to [6].

*Note:* Zero-shot, One-Shot and Few-Shot refer to the number of additional query and answer pairs passed in the context when performing the sampling-based evaluations, not the "shots" involved in hyperparameter transfer.

| Task | Split | Metric | Zero-shot | | | One-shot | | | Few-shot | | |
|---|---|---|---|---|---|---|---|---|---|---|---|
| | | | $\mu P$ | [6] | re-run | $\mu P$ | [6] | re-run | $\mu P$ | [6] | re-run |
| Validation dataset | valid | ce | **1.98** | | 2.03 | | | | | | |
| PTB | test | ppl | **11.4** | | 13.0 | | | | | | |
| Wikitext 103 | test | ppl | **8.56** | | 9.13 | | | | | | |
| LM1B | test | ppl | **20.5** | | 21.7 | | | | | | |
| HellaSwag | dev | acc | **72.0** | 67.4 | 66.7 | **71.1** | 66.5 | 65.9 | **72.4** | 67.3 | 66.4 |
| LAMBADA | test | acc | **73.5** | 70.3 | 70.8 | **69.9** | 65.4 | 64.8 | 74.7 | **79.1** | 77.1 |
| StoryCloze | test | acc | **79.4** | 77.7 | 77.3 | **80.6** | 78.7 | 78.3 | **84.2** | 81.2 | 81.1 |
| NaturalQS | test | acc | **9.86** | 5.79 | 7.20 | **14.7** | 9.78 | 10.6 | **20.2** | 17.0 | 15.7 |
| TriviaQA | dev | acc | **47.0** | 38.7 | 37.5 | **50.4** | 44.4 | 42.5 | **55.5** | 51.6 | 49.9 |
| WebQS | test | acc | **11.3** | 7.73 | 9.79 | **20.2** | 15.1 | 16.2 | **33.0** | 27.7 | 28.2 |
| Ro→En 16 | test | BLEU-sb | **26.9** | 8.75 | 13.7 | **36.5** | 34.2 | 33.5 | **38.2** | 36.2 | 35.6 |
| En→Ro 16 | test | BLEU-sb | **18.1** | 5.31 | 4.40 | **21.0** | 18.2 | 17.3 | **22.0** | 19.6 | 18.8 |
| Fr→En 14 | test | BLEU-sb | **29.8** | 15.5 | 19.6 | **31.7** | 31.6 | 30.1 | **38.0** | 36.4 | 36.5 |
| En→Fr 14 | test | BLEU-sb | **29.6** | 11.4 | 11.6 | **28.8** | 28.3 | 26.0 | 33.3 | **33.3** | 31.2 |
| De→En 16 | test | BLEU-sb | **31.7** | 18.2 | 21.7 | **33.3** | 31.9 | 31.1 | **38.9** | 36.5 | 36.2 |
| En→De 16 | test | BLEU-sb | **23.1** | 9.36 | 9.00 | **24.6** | 21.7 | 21.1 | **27.6** | 24.1 | 24.5 |
| Winograd | test | acc | 85.3 | 85.7 | **86.8** | **84.6** | 84.6 | 84.2 | **86.4** | 85.4 | 83.9 |
| Winogrande | dev | acc | **66.8** | 64.5 | 62.5 | **67.6** | 65.8 | 64.5 | **71.0** | 67.4 | 67.2 |
| PIQA | dev | acc | **79.1** | 78.0 | 78.0 | **77.3** | 76.3 | 76.9 | **79.2** | 77.8 | 77.7 |
| ARC (Challenge) | test | acc | 42.1 | 41.4 | **42.5** | **44.0** | 41.5 | 42.4 | **43.8** | 43.7 | 42.7 |
| ARC (Easy) | test | acc | **64.3** | 60.2 | 61.9 | **65.3** | 62.6 | 63.4 | **67.3** | 65.8 | 65.3 |
| OpenBookQA | test | acc | **54.4** | 50.4 | 52.6 | **56.4** | 53.0 | 52.8 | **58.4** | 55.2 | 54.4 |
| Quac | dev | f1 | **41.8** | 36.1 | 38.2 | **43.1** | 39.0 | 39.5 | **44.0** | 39.9 | 39.9 |
| RACE-h | test | acc | **45.0** | 44.1 | 43.2 | **44.9** | 44.3 | 42.9 | **45.2** | 44.7 | 43.4 |
| RACE-m | test | acc | **58.4** | 54.4 | 54.0 | **57.9** | 54.7 | 53.8 | **58.6** | 55.4 | 55.4 |
| SQuADv2 | dev | f1 | **59.9** | 52.7 | 50.9 | **64.9** | 57.1 | 54.7 | **68.9** | 62.1 | 58.4 |
| CoQA | dev | f1 | **78.5** | 72.8 | 72.9 | **80.9** | 75.1 | 74.4 | **81.3** | 77.3 | 75.4 |
| DROP | dev | f1 | 17.1 | 17.0 | **17.4** | 23.3 | **27.3** | 25.7 | **33.9** | 29.7 | 28.7 |
| BoolQ | dev | acc | **69.4** | 65.4 | 60.9 | **74.1** | 68.7 | 65.0 | **73.9** | 70.0 | 69.7 |
| CB | dev | acc | 21.4 | 28.6 | **37.5** | **60.7** | 33.9 | 32.1 | 62.5 | 60.7 | **66.1** |
| Copa | dev | acc | **82.0** | 80.0 | 77.0 | 81.0 | **82.0** | 81.0 | **88.0** | 83.0 | 82.0 |
| RTE | dev | acc | **55.2** | 55.2 | 46.2 | **61.0** | 54.9 | 58.8 | 52.7 | 49.5 | **59.9** |
| WiC | dev | acc | **0.** | **0.** | **0.** | 50.0 | 50.3 | **50.3** | 50.5 | **53.1** | 51.3 |
| ANLI R1 | test | acc | **33.7** | 32.3 | 33.4 | **32.4** | 31.6 | 31.7 | 30.9 | **33.1** | 30.7 |
| ANLI R2 | test | acc | **33.8** | 33.5 | 33.0 | **34.8** | 33.9 | 33.7 | **35.0** | 33.3 | 32.2 |
| ANLI R3 | test | acc | 32.7 | **34.8** | 33.4 | **34.8** | 33.1 | 33.3 | **36.9** | 33.9 | 32.3 |

Table 9: Evaluation results comparing the GPT-3 6.7B model trained with $\mu$Transfer against the twice as large GPT-3 13B model from [6]. The two models have similar performance on most of the evaluation tasks.

| Task | Split | Metric | Zero-shot | | One-shot | | Few-shot | |
|---|---|---|---|---|---|---|---|---|
| | | | 6.7B+$\mu$P | 13B[6] | 6.7B+$\mu$P | 13B[6] | 6.7B+$\mu$P | 13B[6] |
| HellaSwag | dev | acc | **72.0** | 70.9 | **71.1** | 70.0 | **72.4** | 71.3 |
| LAMBADA | test | acc | **73.5** | 72.5 | **69.9** | 69.0 | 74.7 | **81.3** |
| StoryCloze | test | acc | 79.4 | **79.5** | **80.6** | 79.7 | **84.2** | 83.0 |
| NaturalQS | test | acc | **9.86** | 7.84 | **14.7** | 13.7 | 20.2 | **21.0** |
| TriviaQA | dev | acc | **47.0** | 41.8 | 50.4 | **51.3** | 55.5 | **57.5** |
| WebQS | test | acc | **11.3** | 8.22 | **20.2** | 19.0 | 33.0 | **33.5** |
| Ro→En 16 | test | BLEU-sb | **26.9** | 20.8 | 36.5 | **36.7** | 38.2 | **38.4** |
| En→Ro 16 | test | BLEU-sb | **18.1** | 6.43 | **21.0** | 20.8 | **22.0** | 21.8 |
| Fr→En 14 | test | BLEU-sb | **29.8** | 22.4 | **31.7** | 31.4 | 38.0 | **38.3** |
| En→Fr 14 | test | BLEU-sb | **29.6** | 15.3 | 28.8 | **30.1** | 33.3 | **35.5** |
| De→En 16 | test | BLEU-sb | **31.7** | 24.4 | 33.3 | **34.5** | 38.9 | **39.1** |
| En→De 16 | test | BLEU-sb | **23.1** | 11.0 | **24.6** | 23.3 | 27.6 | **27.7** |
| Winograd | test | acc | 85.3 | **87.9** | 84.6 | **86.1** | **86.4** | 82.4 |
| Winogrande | dev | acc | 66.8 | **67.9** | **67.6** | 66.9 | **71.0** | 70.0 |
| PIQA | dev | acc | **79.1** | 78.5 | 77.3 | **77.8** | 79.2 | **79.9** |
| ARC (Challenge) | test | acc | 42.1 | **43.7** | **44.0** | 43.1 | 43.8 | **44.8** |
| ARC (Easy) | test | acc | **64.3** | 63.8 | 65.3 | **66.8** | 67.3 | **69.1** |
| OpenBookQA | test | acc | 54.4 | **55.6** | **56.4** | 55.8 | 58.4 | **60.8** |
| Quac | dev | f1 | **41.8** | 38.4 | **43.1** | 40.6 | **44.0** | 40.9 |
| RACE-h | test | acc | **45.0** | 44.6 | **44.9** | 44.6 | **45.2** | 45.1 |
| RACE-m | test | acc | **58.4** | 56.7 | **57.9** | 56.9 | **58.6** | 58.1 |
| SQuADv2 | dev | f1 | **59.9** | 56.3 | **64.9** | 61.8 | **68.9** | 67.7 |
| CoQA | dev | f1 | **78.5** | 76.3 | **80.9** | 77.9 | **81.3** | 79.9 |
| DROP | dev | f1 | 17.1 | **24.0** | 23.3 | **29.2** | **33.9** | 32.3 |
| BoolQ | dev | acc | **69.4** | 66.2 | **74.1** | 69.0 | **73.9** | 70.2 |
| CB | dev | acc | **21.4** | 19.6 | **60.7** | 55.4 | 62.5 | **66.1** |
| Copa | dev | acc | 82.0 | **84.0** | 81.0 | **86.0** | **88.0** | 86.0 |
| RTE | dev | acc | 55.2 | **62.8** | **61.0** | 56.3 | 52.7 | **60.6** |
| WiC | dev | acc | **0.** | **0.** | **50.0** | **50.0** | 50.5 | **51.1** |
| ANLI R1 | test | acc | **33.7** | 33.2 | 32.4 | **32.7** | 30.9 | **33.3** |
| ANLI R2 | test | acc | **33.8** | 33.5 | **34.8** | 33.9 | **35.0** | 32.6 |
| ANLI R3 | test | acc | 32.7 | **34.4** | **34.8** | 32.5 | **36.9** | 34.5 |

conduct a learning rate sweep for models of different widths using SP; the result is shown in Fig. 16, on the left.

**Hyperparameter Stability**    Note that the best model with a width multiplier of 8 under-performs that with a multiplier of 4. We run the same sweep with $\mu$P, along with a sweep of the output multiplier ($\alpha_{output}$); the result is shown in Fig. 16, on the right. We notice that wider models always perform better under $\mu$P and that the optimal learning rate $\eta$ and $\alpha_{output}$ are stable across width.

**Hyperparameter Transfer**    Next, we perform a grid search for learning rate ($\eta$) and $\alpha_{output}$ on the 0.5x model for both SP and $\mu$P.[26] Then, we take the best combination and test on the 8x model, simulating how a practitioner might use $\mu$Transfer. The result is shown in Table 10, where $\mu$P outperforms SP by $0.43\% \pm .001\%$.

### I.1.2   Wide ResNet on ImageNet

**Setup**    For this case we use Wide-Resnet, or WRN [50], a ResNet variant with more channels per layer, to further showcase $\mu$Transfer across width, i.e., number of channels. We train with SGD on ImageNet for 50 epochs following standard data augmentation procedures. We use a width multiplier to identify models of different width, and a multiplier of 1 corresponds to the original WRN-50-2-bottleneck in [50].

---

[26]Here we tune the 0.5x model instead of the 1x model to simulate the situation that one does "exploratory work" on the 1x model but, when scaling up, would like to tune faster by using a smaller proxy model.

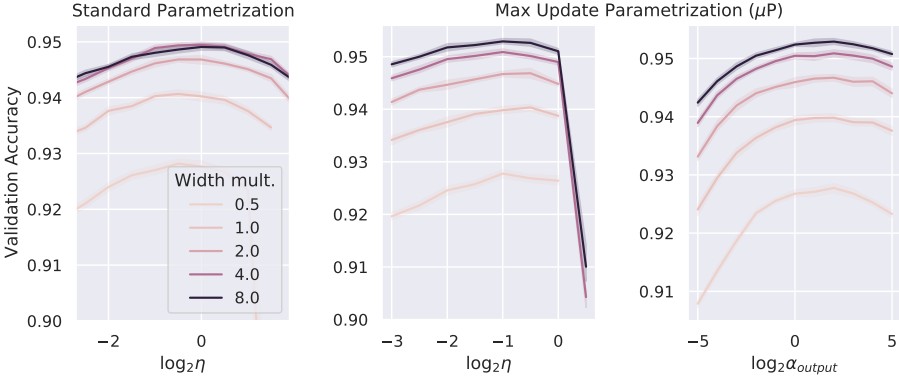

Figure 16: ResNet on CIFAR-10 for different widths (compared to a base network). On the **left**, the widest network SP underperforms; on the **right**, the $\mu$P network has a more consistent HP landscape and performs better. Both networks are tuned at the smallest width for the HP ($\eta$ or $\alpha_{output}$) not in the x-axis.

Table 10: CIFAR10: Transferring the best learning rate ($\eta$) and $\alpha_{output}$ from widening factor $0.5$ to $8$; $\mu$P significantly outperforms SP given the same search grid. The best HPs are different as the models are parametrized to be identical at 1x width.[26]

| Transfer Setup | Best $\eta$ | Best $\alpha_{output}$ | Valid. Acc. (0.5x) | Valid. Acc. (8x) |
|:---:|:---:|:---:|:---:|:---:|
| SP | 0.707 | 4 | 92.82% | 94.86% |
| $\mu$P | 0.5 | 4 | 92.78% | **95.29%** |

**Hyperparameter Transfer**    We start with a proxy model with a width multiplier of 0.125 and tune several HPs using the following grid:

- $\eta$: $1 \times 2.048 \times 2^z$, where $z \in \{-5, -4, -3, ..., 4\}$

- $\alpha_{output}$: $10 \times 2^z$, where $z \in \{-5, -4, -3, ..., 4\}$

- weight decay co-efficient $\gamma$: $3.05 \times 10^{-5} \times 2^z$, where $z \in \{-2, -1.5, -1, ..., 1.5\}$

- SGD momentum $\beta$: $0.875 \times 2^z$, where $z \in \{-2, -1.5, -1, ..., 1.5\}$

The grid is centered around the default HPs used by [1] for ResNet-50; while not expected to be competitive for WRN, they represent a reasonable starting point for our experiment.

We randomly sample 64 HP combinations from the grid and train for 50 epochs, before selecting the one with the highest top-1 validation accuracy. Then, we scale up the model following both $\mu$P and SP and run with the same HPs we just selected. The result is shown in Table 11, where $\mu$P outperforms SP by $0.41\%$ in terms of top-1 validation accuracy.

Table 11: Imagenet: Transferring the best learning rate ($\eta$), $\alpha_{output}$, $\gamma$, and $\beta$ from widening factor $0.125$ to $1$; $\mu$P significantly outperforms SP given the same search grid.

| Transfer Setup | Best $\eta$ | Best $\alpha_{output}$ | Best $\gamma$ | Best $\beta$ | Valid. Acc. (0.125x) | Valid. Acc. (1x) |
|:---:|:---:|:---:|:---:|:---:|:---:|:---:|
| SP | 32.768 | .625 | .000015 | .4375 | 58.12% | 76.75% |
| $\mu$P | 32.768 | .625 | .000015 | .4375 | 58.12% | **77.16%** |

## I.2    Experiments on Transformers

### I.2.1    Verifying Transfer across Batch Size, Sequence Length, and Training Time on Wikitext-2

See Fig. 19.

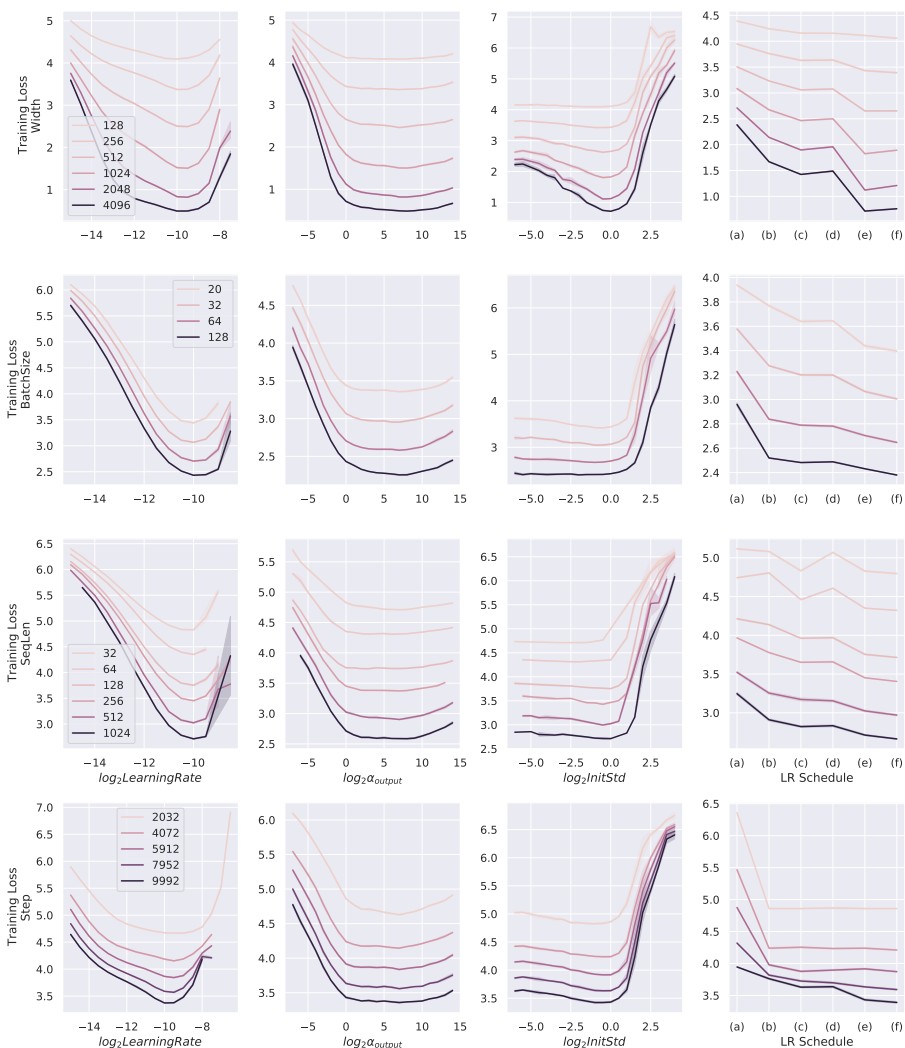

Figure 17: **Empirical validation of $\mu$Transfer for Post-LN Transformers.** Same setting as Fig. 4.

## I.3 Post-Layernorm Transformers

Fig. 17 shows the transferability of learning rate, $\alpha_{output}$, initialization standard deviation, and Adam $\beta_2$ across width, batch size, sequence length, and training steps for post-layernorm Transformers. However, in general, we find transfer across depth to be fragile.

### I.3.1 Hyperparameter Instability of SP Transformers

Fig. 18 and Fig. 20 show the HP instability inherent in SP Transformers.

## J Implementing $\mu$Transfer in a Jiffy

As we have shown, one can enable $\mu$Transfer by just reparametrizing the desired model in Maximal Update Parametrization ($\mu$P). While conceptually simple, switching from Standard Parametrization (SP) to $\mu$P can be error-prone, as popular deep learning frameworks are built around SP. We strive to build a tool that fulfills two goals:

1. Minimize code changes when switching to $\mu$P;
2. Keep model behavior invariant, under this switch, at a given *base model shape*.

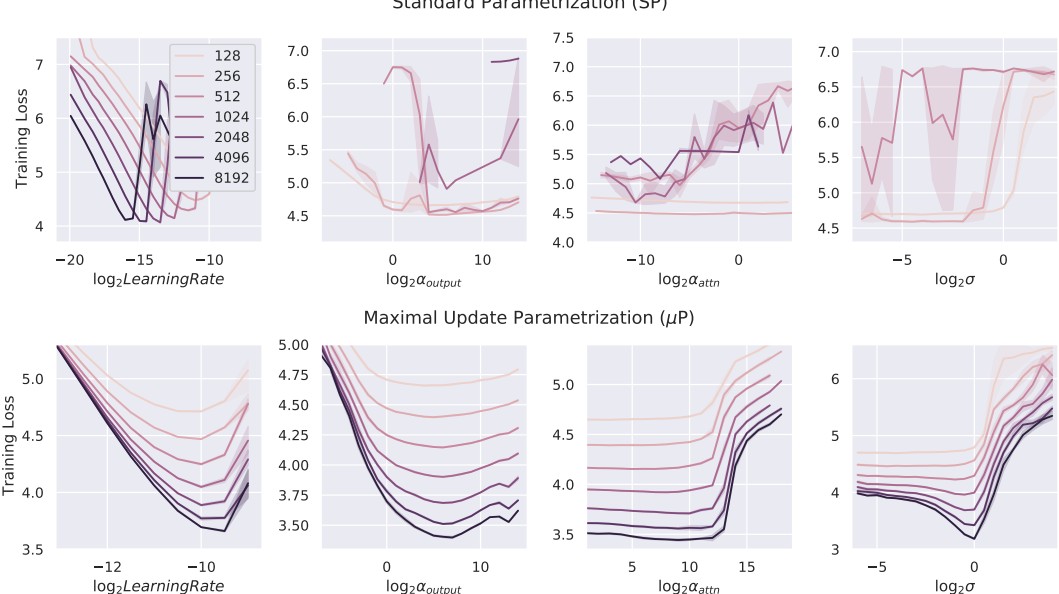

Figure 18: Post-layernorm Transformer with SP and $\mu$P on Wikitext-2. We sweep one HP across width ($d_{model}$) at a time while keeping the rest fixed; we also scale $d_{head}$ linearly with $d_{model}$ and fixing $n_{head}$. $\alpha_{output}, \alpha_{attn}$ are multipliers for output and key weights, and $\sigma$ is initialization standard deviation. This yields unstable result for SP, as expected, where missing points/curves represent divergence; in $\mu$P, the optimal HP choices stabilize as width increases.

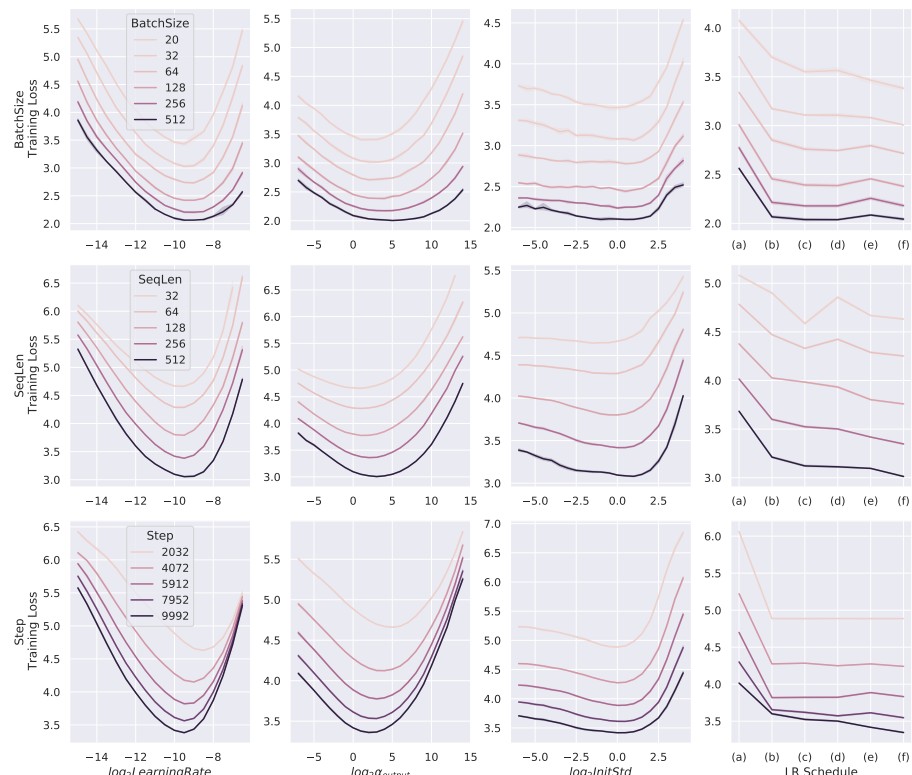

Figure 19: **Empirical validation of Hyperparameter Transfer across Batch Size, Sequence Length, and Training Time on pre-LN Transformers.** Same setting as Fig. 4. Despite some shift, the optimal HPs are roughly stable when transferring from batch size 32, sequence length 128, and 5000 training steps.

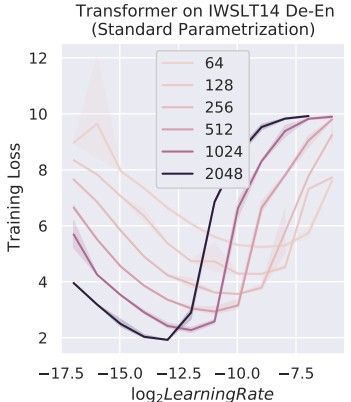

Figure 20: Learning rate landscape is highly unstable under standard parametrization in IWSLT.

Table 12: **Alternative (Equivalent) $\mu$P Formulation for Easier Implementation.** Same format as in Table 3. In contrast to the formulation in Table 3, here all "vector-like" parameters (i.e. those that have only one dimension tending to infinity), including input and output weights and biases, have the same width scaling for initialization variance and SGD/Adam LR (note the $1/\text{fan\_in}$ for input weight/bias init. var. is $\Theta(1)$ in width). This has two benefits in practice: 1) implementation is unified and simplified for all "vector-like" parameters; 2) input and output weights can now be tied, in contrast to Table 3, which is a common design feature of Transformer models. Note that in this table, for biases, the $\text{fan\_in}$ is 1 (compare to PyTorch `nn.Linear` default initialization of biases, where $\text{fan\_in}$ refers to $\text{fan\_in}$ of the layer.)

| | Input weights & all biases | | Output weights | | Hidden weights | |
|---|---|---|---|---|---|---|
| Init. Var. | $1/\text{fan\_in}$ | | 1 | $(1/\text{fan\_in})$ | $1/\text{fan\_in}$ | |
| Multiplier | 1 | | $1/\text{fan\_in}$ | $(1)$ | 1 | |
| SGD LR | $\eta \cdot \text{fan\_out}$ | $(\eta)$ | $\eta \cdot \text{fan\_in}$ | $(\eta)$ | $\eta$ | |
| Adam LR | $\eta$ | | $\eta$ | | $\eta/\text{fan\_in}$ | $(\eta)$ |

By *model shape*, we mean the collection of dimensions of all parameters of the model. The latter goal, which we call *parametrization backward compatibility*, ensures that any code base works exactly as before at the base model shape, similar to Eq. (4), e.g. the loss at any time step remains exactly the same before and after the switch to $\mu$P. Of course, when widths start to differ from the base model shape, the model behavior necessarily changes so that HPs can be transferred.

There are two common approaches to setting the base model shape: 1) If one intends to tune a large target model, then the user can set the base model shape to be the shape of the target model (e.g. BERT-large or T5-large), so that the target model itself is in standard parametrization. Then one can tune a proxy model with e.g. $width = 124$ to obtain the optimal HPs for the target model. In addition, if one wishes to scale up further e.g. $width = 1024$, then these HPs remain optimal. 2) If one has done exploration on a new idea with a small model and now wishes to scale up, reusing the HP found during this exploration, then one can set the base model shape to be the shape of the exploratory small model. Of course, in both scenarios, depth, batch size, and sequence lengths can be scaled up and down as well according to Fig. 19 (though note that currently we require users to recreate the base model shape at new depths, since the number of parameters now change with depth).

**The `mup` Package** We provide our tool as a Python package called `mup` designed to work with PyTorch. The following example illustrates the usage of our package.

Table 13: $\mu$P Formulation in the Style of [45].

|  | Input weights & all biases | | Output weights | | Hidden weights | |
|---|---|---|---|---|---|---|
| Init. Var. | $1/\text{fan\_out}$ | $(1/\text{fan\_in})$ | $1/\text{fan\_in}$ | | $1/\text{fan\_in}$ | |
| Multiplier | $\sqrt{\text{fan\_out}}$ | $(1)$ | $1/\sqrt{\text{fan\_in}}$ | $(1)$ | $1$ | |
| SGD LR | $\eta$ | | $\eta$ | | $\eta$ | |
| Adam LR | $\eta/\sqrt{\text{fan\_out}}$ | $(\eta)$ | $\eta/\sqrt{\text{fan\_in}}$ | $(\eta)$ | $\eta/\text{fan\_in}$ | $(\eta)$ |

```python
from mup.layers import MuReadout
from mup.shape import save_shapes, set_base_shapes
from mup.optim import MuSGD, MuAdam

class MyModel(nn.Module):
    def __init__(self, width, ...):
        ...
        ### In model definition, replace output layer with MuReadout
        # readout = nn.Linear(width, d_out)
        readout = MuReadout(width, d_out)
        ...
    def forward(self, ...):
        ...
        ### If using a transformer, make sure to use
        ### 1/d instead of 1/sqrt(d) attention scaling
        # attention_scores = query @ key.T / d**0.5
        attention_scores = query @ key.T / d
        ...

### Instantiate a base model
base_model = MyModel(width=1)

### Instantiate the target model (the model you actually want to train).
### This should be the same as the base model except the widths could be potentially different.
### In particular, base_model and model should have the same depth
model = MyModel(width=100)

### Set base shapes
set_base_shape(model, base_model)

### Alternatively, one can save the base model shapes in a file
# save_shapes(base_model, filename)
### and later set base shapes directly from the filename
# set_base_shape(model, filename)
### This is useful when one cannot fit both base_model and model in memory at the same time

for param in model.parameters():
    ### If initializing manually with fixed std or bounds,
    ### then replace with same function from mup.init
    # torch.nn.init.uniform_(param, -0.1, 0.1)
    mup.init.uniform_(param, -0.1, 0.1)
    ### Likewise, if using
    ### `xavier_uniform_, xavier_normal_, kaiming_uniform_, kaiming_normal_`
    ### from `torch.nn.init`, replace with the same functions from `mup.init`

### Use the optimizers from `mup.optim` instead of `torch.optim`
# optimizer = torch.optim.SGD(model.parameters(), lr=0.1)
optimizer = MuSGD(model.parameters(), lr=0.1)

### Then just train normally
```

**What Happens in the** `mup` **Package** Under the hood, `mup` implements the $\mu$P formulation in Table 12. By invoking `set_base_shape(model, base_model)`, each parameter tensor `p` of `model` gets a `p.infshape` attribute that stores, for each of its dimensions, the corresponding base dimension and whether that dimension should be considered "infinite" (i.e. will be scaled up/down, e.g., $d_{model}$ of a Transformer) or "finite" (i.e. will be fixed, e.g., vocabulary size). This information is used in the initializers and optimizers to automatically scale the parameters or learning rates to be compliant with $\mu$P. For example, by Table 12, the Adam learning rate of hidden weights `p` is calculated as $\eta/\text{p.infshape.width\_mult()}$, where `p.infshape.width_mult()` essentially calculates $\frac{\text{fan\_in}}{\text{base\_fan\_in}}$.

# K   Reverse-$\mu$Transfer for Diagnosing Training Instability in Large Models

Large Transformers are famously fickle to train [20, 29]. We note that a possible source of this instability for larger transformers is the failure of naive hyperparameter transfer via the standard parametrization. This is certainly consistent with Fig. 1, which shows that the optimal learning rate for small Transformers can lead to trivial performance in large Transformers. We support this

hypothesis further by *reverse-μTransferring the instability-inducing HPs from a large Transformer to a small one and replicating the training instability*. This is shown in Fig. 21.

Practically, this reverse-μTransfer technique can be used to diagnose or debug training instability problems of large models. We offer two case studies toward this claim.

1) When training transformers of width 8192 on Wikitext-2, we found certain HP combinations caused divergence in the middle of training. We reverse-μTransferred one such HP combination to a model of width 256 and replicated this divergence. By analyzing this small model's activations right before this divergence, we found that the cause is due to attention logits blowing up. Note this debugging session proceeded much more quickly than if we directly worked with the large model. Later we confirmed this is indeed the same cause of the width-8192 model's divergence.

2) A 6B-parameter language model (in standard parametrization) in a separate project experienced repeated blow-up in the middle of training. We reverse-μTransferred its hyperparameters to a smaller, 100M-parameter model and replicated the training instability. This was solved by a retuning of the small model via random search.

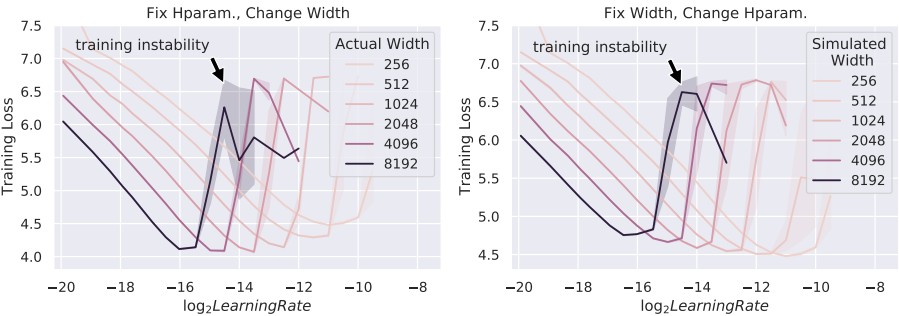

Figure 21: **Replicating training instability on a small Transformer by *reverse-μTransferring* hyperparameters.** These experiments concern 2-layer Transformers in Standard Parametrization (SP) on Wikitext-2, trained with Adam, where width is defined as $d_{model} = d_{ffn}$. (Left) LR-vs-loss for wider and wider Transformers. (Right) Likewise for *simulated width*: Here each point $(\log_2 \eta, loss)$ for simulated width $n$ indicates the loss from training a width-256 μP Transformer with base width $n$ and LR $\eta$ (i.e. loosely speaking, it's using LR transferred from $\eta$ in a width-$n$ SP Transformer). **Takeaway:** The overall shapes of the curves are identical between the left and right plots[27]; in particular, a learning rate leads to instability in a wide model iff it does so when transferred back to a narrow model.

## L  An Intuitive Introduction to the Theory of Maximal Update Parametrization

In what follows, we seek to describe useful intuitions and rule of thumbs that would be helpful to practitioners and empirical researchers alike in figuring out what is the right neural network parametrization. The intuitions we shall describe regarding SGD can be made rigorous as in [44, 45]; those regarding Adam are new, and their formalization will be done in an upcoming paper.

### L.1  Behaviors of Gaussian Matrices vs Tensor Product Matrices

Central to the derivation of μP for any architecture are key insights on the behaviors of two kinds of random matrices: 1) iid Gaussian random matrix and 2) tensor product matrix (by which we mean a sum of outer products) and more generally what we call *nonlinear* tensor product matrix (see Eq. (7)). For example, a neural network, randomly initialized in the typical way, will have each weight matrix look like the former. However, every step of training by gradient descent adds a sum of outer products to this initial matrix, so that the *change in weights* constitute a tensor product matrix. For Adam, the change in weights is not a tensor product but a more general *nonlinear tensor product matrix (see Eq. (7))*. In this section, we will particularly focus on the *right scaling* for the entries of such

---

[27] Note that the curves on the left are "lower" than curves on the right. This just reflects the increasing capacity of wider models able to fit the training data better, so is orthogonal to our point.

Table 14: Expected entry size of $Av$ for different matrices $A$ and vector $v$ correlated with each other, both having entries of size $\Theta(1)$.

| | Standard Gaussian $A \in \mathbb{R}^{n \times n}$ | (Nonlinear) Tensor Product $A \in \mathbb{R}^{n \times n}$ | Vector $A \in \mathbb{R}^{1 \times n}$ |
|---|---|---|---|
| Entry size of $Av$ | $\Theta(\sqrt{n})$ | $\Theta(n)$ | $\Theta(n)$ |

matrices, leading to a discussion of *the right neural network parametrization* in the next section. We concentrate on the key heuristics but eschew burdensome rigor.

**Key Insights** Consider a random vector $v \in \mathbb{R}^n$ with approximately iid entries and a random matrix $A$ of either size $n \times n$ or $1 \times n$, both having entries of size $\Theta(1)$.[28] In the context of deep learning, $v$ for example can be an activation vector in an MLP, a Gaussian $A$ the hidden weights at initialization, a (nonlinear) tensor product $A$ the change in hidden weights due to training, and a vector $A$ the readout layer weights. Then $Av$ corresponds to a part of the next layer preactivation or the network output. To make sure the preactivations and the output don't blow up, we thus need to understand the scale of $Av$, especially in the general case where $A$ is correlated with $v$.[29] This is summarized in Table 14, with the derivations below. Intuitively, a (nonlinear) tensor product or vector $A$ will interact with a correlated $v$ via Law of Large Numbers, hence the $n$-scaling, while a Gaussian $A$ interacts with $v$ via Central Limit Theorem, hence the $\sqrt{n}$-scaling.

In the derivations below, we answer a slightly different but equivalent question of "how to scale $A$ such that $Av$ has entry size $\Theta(1)$?"

### L.1.1 Preparation for the Derivations

By the results of [45], each (pre-)activation vector and its gradient vector in a multi-layer perceptron have approximately iid coordinates in the large width limit,[30] and something similar can be said for more advanced networks such as ResNet and Transformers [31]. In particular, to each such vector $v$, we can associate a random variable $Z^v$ that represents the coordinate distribution of $v$. If vector $u$ is correlated with $v$, then $Z^u$ will also be correlated with $Z^v$, and $\lim_{n \to \infty} v^\top u / n = \mathbb{E} Z^u Z^v$.

### L.1.2 Linear Tensor Product Matrix (e.g. SGD Updates)

The case of (linear) tensor product matrix can be reduced to the outer product case by linearity. Given $u, v, x \in \mathbb{R}^n$ having approximately iid coordinates (of size $\Theta(1)$) like so, we can form the outer product

$$A \stackrel{\text{def}}{=} u \otimes v/n = uv^\top/n, \tag{6}$$

which is the form of a single (batch size 1) gradient update to a weight matrix. Then, by Law of Large Numbers,

$$Ax = u\frac{v^\top x}{n} \approx cu, \quad \text{where} \quad c = \mathbb{E} Z^v Z^x.$$

So $Ax$ also has approximately iid coordinates, distributed like $Z^{Ax} \stackrel{\text{def}}{=} Z^u \mathbb{E} Z^v Z^x$. Likewise, if $A$ is a sum of outer products $A = \sum_{i=1}^k u^i \otimes v^i/n$, then

$$Ax = \sum_{i=1}^k u^i \frac{v^{i\top} x}{n}, \quad \text{with coordinates distributed as} \quad Z^{Ax} = \sum_{i=1}^k Z^{u^i} \mathbb{E} Z^{v^i} Z^x.$$

Notice that each coordinate of $A$ has size $\Theta(1/n)$. The above reasoning shows that, in order for $Ax$ to have coordinate size $\Theta(1)$ (assuming $x$ does), then $\Theta(1/n)$ is the right coordinate size for $A$, in

---

[28]in the sense that the the variance of the entries are $\Theta(1)$

[29]Here "correlated" formally means $v$ depends on $W^\top$ in a Tensor Program. This essentially captures all scenarios of "$v$ correlated with $W$" that occurs in deep learning.

[30]Our intuition here is derived from the assumption that width is much larger than training time; of course, as illustrated by our myriad experiments, these intuition are very useful even when this is not the case, such as when training to convergence.

[31]E.g. in a convnet, the (pre-)activations are iid across channels, but correlated across pixels

the general case that $v^i$ and $x$ are correlated (as is generically the case during gradient descent, with $A = \Delta W$ for some weights $W$ and $x$ being the previous activations).[32]

### L.1.3 Nonlinear Tensor Product Matrix (e.g. Adam Updates)

When using Adam or another adaptive optimizer that normalizes the gradient coordinatewise before applying them, we need to modify our argument slightly to obtain the right coordinate size scaling of the matrix. The gradient update $A$, after such normalization, will take the form of

$$A_{\alpha\beta} = \psi(u^1_\alpha, \ldots, u^k_\alpha, v^1_\beta, \ldots, v^k_\beta), \quad \text{for some } \psi : \mathbb{R}^{2k} \to \mathbb{R} \text{ and vectors } u^i, v^j \in \mathbb{R}^n. \quad (7)$$

We say a matrix of this form is a *nonlinear tensor product matrix.*

First, note the tensor product matrices (e.g. the form of SGD update) discussed previously (Eq. (6)) already takes this form, with $\psi(u^1_\alpha, \ldots, u^k_\alpha, v^1_\beta, \ldots, v^k_\beta) = n^{-1}(u^1_\alpha v^1_\beta + \cdots + u^k_\alpha v^k_\beta)$, so Eq. (7) is a strict generalization of linear tensor products. Next, for the example of Adam, each gradient update is $\mu/\sigma$ where $\mu$ (resp. $\sigma^2$) is the moving average of previous (unnormalized) gradients (resp. the coordinatewise square of the same).[33] If these unnormalized gradients are the outer products $u^1 \otimes v^1, \ldots, u^k \otimes v^k$, then the update has coordinates

$$(\mu/\sigma)_{\alpha\beta} = \psi(u^1_\alpha, \ldots, u^k_\alpha, v^1_\beta, \ldots, v^k_\beta) \stackrel{\text{def}}{=} \sum_i \gamma_i u^i_\alpha v^i_\beta \Big/ \sqrt{\sum_i \omega_i (u^i_\alpha v^i_\beta)^2}, \quad (8)$$

where $\gamma_i$ and $\omega_i$ are the weights involved in the moving averages.

Now suppose we have some $A \in \mathbb{R}^{n \times n}$ of the form Eq. (7), where $u^i, v^i \in \mathbb{R}^n$ have approximately iid coordinates (of size $\Theta(1)$), and $\psi = n^{-1}\bar\psi$ where $\bar\psi$ doesn't depend on $n$ (in terms of Adam where $\bar\psi$ corresponds to the $\psi$ of Eq. (8), this corresponds to using a learning rate of $1/n$). Then for $x \in \mathbb{R}^n$ having approximately iid coordinates of size $\Theta(1)$, by Law of Large Numbers,

$$(Ax)_\alpha = \frac{1}{n} \sum_{\beta=1}^n \bar\psi(u^1_\alpha, \ldots, u^k_\alpha, v^1_\beta, \ldots, v^k_\beta) x_\beta \approx \mathbb{E}\, \bar\psi(u^1_\alpha, \ldots, u^k_\alpha, Z^{v^1}, \ldots, Z^{v^k}) Z^x \stackrel{\text{def}}{=} \Psi(u^1_\alpha, \ldots, u^k_\alpha).$$

Here we made the obvious definition

$$\Psi : \mathbb{R}^k \to \mathbb{R}, \qquad \Psi(r_1, \ldots, r_k) \stackrel{\text{def}}{=} \mathbb{E}\, \bar\psi(r_1, \ldots, r_k, Z^{v^1}, \ldots, Z^{v^k}) Z^x.$$

Thus $Ax$ also has approximately iid coordinates (of size $\Theta(1)$),

$$Z^{Ax} \stackrel{\text{def}}{=} \Psi(Z^{u^1}, \ldots, Z^{u^k}).$$

For example, in the SGD example with $A = u \otimes v/n$ and $\bar\psi(u_\alpha, v_\beta) = u_\alpha v_\beta$, this formula gives $Z^{Ax} = \Psi(Z^u)$ where $\Psi(z) = z\, \mathbb{E}\, Z^v Z^x$, recovering the earlier derivation.

In any case, the point here is that $A$ has coordinate size $\Theta(1/n)$, and this is the unique scaling that leads to $Ax$ having coordinate size $\Theta(1)$.

### L.1.4 Vector Case (e.g. Readout Layer)

The vector $A$ case is similar to the tensor product cases above.

### L.1.5 Gaussian Matrix (e.g. Hidden Weights Initialization)

Now consider the case where $A \in \mathbb{R}^{n \times n}$ is random Gaussian matrix with $A_{\alpha\beta} \sim \mathcal{N}(0, 1/n)$ and $x \in \mathbb{R}^n$ has approximately iid coordinates distributed like $Z^x$. In the context of neural network training, $A$ should be thought of as a randomly initialized weight matrix, and $x$ for example can be taken to be an activation vector in the first forward pass.

---

[32]In some corner cases when $x$ is uncorrelated with $v$, then $v^\top x = \Theta(\sqrt{n})$ by Central Limit, so actually $Ax$ has $\Theta(1/\sqrt{n})$ coordinates. However, this case does not come up much in the context of training neural networks.

[33]Adam also has bias correction for the moving averages which can be accomodated easily, but for simplicity we omit them here.

**A Quick Intuition**  By standard random matrix theory, $A$ has $\Theta(1)$ operator norm with high probability. Thus, with high probability, for *any vector* $x$, we have $\|Ax\| \approx \|x\|$, even if $x$ is correlated with $A$. If $Ax$'s coordinates are "evenly distributed", then this would imply $Ax$ has $\Theta(1)$ coordinates if $x$ does. However, this is not so clear. Below we provide intuitions for why this would be the case.

**Intuition for Evenness of Coordinate Distribution**  If $x$ is independent from $A$ (or sufficiently uncorrelated), then each coordinate $(Ax)_\alpha$ has variance $\mathbb{E}(Z^x)^2 = \Theta(1)$ (so by definition has size $\Theta(1)$). Thus, here $A$ having $\Theta(1/\sqrt{n})$ coordinates leads to $Ax$ having $\Theta(1)$ coordinates, in contrast to the tensor product case above.

When $x$ is correlated with $A$, it turns out the same scaling applies ($\Theta(1/\sqrt{n})$ is the unique scaling for $A$'s entries such so that $Ax$ has $\Theta(1)$ entries), but the reasoning is much more subtle: In the context of neural network training, it turns out all scenario where $x$ is correlated with $A$ can be reduced to the case where $x = \phi(A^\top y, \ldots)$ for some coordinatewise nonlinearity $\phi$ and some other vector $\mathbb{R}^n$.[34]  Let's consider a very simple example with $x = A^\top \mathbf{1}$ for the all 1s vector $\mathbf{1} \in \mathbb{R}^n$ (which has coordinate size $\Theta(1)$ as can be checked easily). Then, for each index $\alpha \in [n]$, we can calculate

$$(AA^\top \mathbf{1})_\alpha = \sum_{\beta,\gamma} A_{\alpha\beta} A_{\gamma\beta} = \sum_\beta A_{\alpha\beta}^2 + \sum_\beta \sum_{\gamma \neq \alpha} A_{\alpha\beta} A_{\gamma\beta}.$$

Since $\mathbb{E} A_{\alpha\beta}^2 = 1/n$, by the Law of Large Number, the first sum $\sum_\beta A_{\alpha\beta}^2 \approx 1$. On the other hand, there are $n$ summands of the form $\sum_{\gamma \neq \alpha} A_{\alpha\beta} A_{\gamma\beta}$, all iid with variance $\frac{n-1}{n^2} = \Theta(1/n)$. Thus by the Central Limit Theorem, we expect $\sum_\beta \sum_{\gamma \neq \alpha} A_{\alpha\beta} A_{\gamma\beta} \approx \mathcal{N}(0,1)$. Therefore, each coordinate of $(AA^\top \mathbf{1})_\alpha$ looks like $1 + \mathcal{N}(0,1) = \mathcal{N}(1,1)$ and thus has size $\Theta(1)$; again this is caused by $A$ having $\Theta(1/\sqrt{n})$ coordinates.

This example can be generalized to more general $x$ that is correlated with $A$, but the mathematics is quite involved. See [44] for more details.

## L.2  Deriving $\mu$P for Any Architecture

Armed with the insight from the last section, we now outline the key steps to derive $\mu$P in Table 3 for any architecture. In practice, $\mu$P implies the following desiderata

**Desiderata L.1.**  At any time during training

1. Every (pre)activation vector in a network should have $\Theta(1)$-sized coordinates[35]

2. Neural network output should be $O(1)$.

3. All parameters should be updated as much as possible (in terms of scaling in width) without leading to divergence.

Let's briefly justify these desiderata. For the desideratum 1, if the coordinates are $\omega(1)$ or $o(1)$, then for sufficiently wide networks their values will go out of floating point range. This problem is particularly acute for low-precision formats that are essential for training large models such as BERT or GPT. Moreover, a general nonlinearity is only well-behaved if its input is in a fixed range (although this is not a problem for homogeneous nonlinearities like relu). For example, for tanh nonlinearity, if the preactivation is vanishing $o(1)$, then $\tanh$ is essentially linear; if the preactivation is exploding $\omega(1)$, then the tanh gradient vanishes.

For the desideratum 2, a similar justification applies to the numerical fidelity of the loss function and loss derivative. Note that, with desideratum 3, this means the network output should be $\Theta(1)$ after training (but it can go to zero at initialization).

Finally, desideratum 3 means that 1) we are doing "maximal feature learning" [45] and 2) every parameter contribute meaningfully in the infinite-width limit. This ensures that learning rate "plays the same role" in the finite-width case as in the infinite-width limit. For example, it prevents the scenario where a weight matrix gets stuck at initialization in the limit for any learning rate (so

---

[34]This is because every "reasonable" deep learning computation can be expressed in a Tensor Program.

[35]In a convnet, a (pre-)activation vector corresponds to a single pixel across all channels; in general , we expect (pre-)activations are iid across channels, but correlated across pixels

learning rate does not matter) but evolves nontrivially in any finite-width network (so learning rate does matter).

These desiderata will essentially uniquely single out $\mu$P. More formally, $\mu$P is the unique parametrization that admits feature learning in all parameters of the neural network [45], and this property theoretically guarantees HP transfer across width (for sufficiently large width). However, for the sake of reaching a broader audience, we will focus more on the intuitive derivations from the desiderata rather than on this formal aspect.

Below, we first assume for simplicity that the width of every layer is $n$, and we focus only on dense weights. Later, we will discuss convolutions and varying the widths between layers.

### L.2.1   $\mu$P Derivation From the Desiderata

**Output Weights**   Suppose $W \in \mathbb{R}^{1 \times n}$ is an output weight. By desideratum 1, the input $x$ to $W$ has $\Theta(1)$-sized coordinates. Thus $W$ should have $\Theta(1/n)$ coordinates so that $Wx = O(1)$. We can initialize $W$ with $\Theta(1/n)$ coordinates and scale its (per-layer) LR so that $\Delta W$ has $\Theta(1/n)$ coordinates as well. This means initializing $W_{\alpha\beta} \sim \mathcal{N}(0, \Theta(1/n^2))$ and use $\Theta(1/n)$ learning rate for both SGD and Adam.

**Hidden Weights**   Consider a square weight matrix $W \in \mathbb{R}^{n \times n}$. Desiderata 1 guarantees that the input $x$ to $W$ has $\Theta(1)$-sized coordinates. Generally, $x$ will be correlated with $W$. By Table 14, we can immediately derive

*Initialization* $W$ should be randomly initialized with coordinate size $\Theta(1/\sqrt{n})$

*LR* The learning rate should be scaled so that $\Delta W$ has coordinate size $\Theta(1/n)$

so that $(W_0 + \Delta W)x$ is $\Theta(1)$ if $x$ is, inductively satisfying desideratum 1. With Adam, this just means the per-layer LR is $\Theta(1/n)$. With SGD and the scaling of output layers above, we can calculate that the gradient of $W$ has $\Theta(1/n)$ coordinates, so the $\Theta(1)$ SGD LR derived above suffices as well.

**Input Weights**   Suppose $W \in \mathbb{R}^{n \times d}$ is an input weight. To satisfy desideratum 1 (i.e. for any input $\xi$, $W\xi$ should have $\Theta(1)$ coordinates), we want $W$ to have $\Theta(1)$ coordinates. We can initialize $W$ with $\Theta(1)$ coordinates and scale its (per-layer) LR so that $\Delta W$ has $\Theta(1)$ coordinates as well. This implies initialization variance of $\Theta(1)$ (or $\Theta(1/\text{fan\_in})$ since $\text{fan\_in} = \Theta(1)$ here) and Adam learning rate $\Theta(n)$. As above, we can calculate that the gradient of $W$ has $\Theta(1/n)$ coordinates, so we want SGD learning rate $\Theta(n)$.

**Biases**   Biases follow the same reasoning as input weights (just think of it as an input weight with input 1).

**Attention**   Suppose the key dimension $d_k$ is tending to infinity with width with number of heads $n_{head}$ fixed. Then the key-query contraction $q^\top k \in \mathbb{R}$ scales like $\Theta(d_k)$ by Law of Large Numbers (instead of Central Limit Theorem because $q$ and $k$ are generally correlated) and desideratum 1, hence the $1/d_k$ we propose rather than $1/\sqrt{d_k}$.

Now suppose instead that $n_{head}$ tends to infinity with width with $d_k$ fixed. Let $K, Q \in \mathbb{R}^{N \times d_k \times n_{head}}, V \in \mathbb{R}^{N \times d_v \times n_{head}}$ be keys, queries, and values across all heads and tokens. Thinking of $N \times d_k$ as constants, we may view attention as a nonlinearity coordinatewise in the $n_{head}$ dimension. Then it's clear that our parametrization described above already works.

Finally, we may freely let $d_k$ and $n_{head}$ both tend to infinity, and the above reasoning shows that our parametrization still works.

**Changing Width Ratios**   As noted above, at any time in training, every (pre-)activation vector will have approximately iid coordinates (of order $\Theta(1)$ by desideratum 1). Another desideratum for $\mu$P is to ensure that this coordinate distribution (at any particular time) stays roughly invariant as widths increases. When all layer widths are tied, this is automatic if the other desiderata are satisfied, hence why we did not list this above.

When width ratios vary, this is not automatic. In this case, we need to choose whether to replace each $n$ with fan-in or fan-out (or some function of them). Making the wrong choices will let the coordinate distributions vary with width ratios.

Obviously, we should replace $n$ with fan-in for the output layers and with fan-out for the input layers since they are the only dimension scaling with $n$. For the hidden weights, we replace $n$ with fan-in so that the forward pass is preserved. When using Adam (and assuming the initialization of $W$ is quickly dominated by the change in $W$), this ensures that the (pre-)activation coordinate distributions are preserved at any time during training even if we vary widths in different layers differently. (For SGD this doesn't quite work in general because the varying width ratios change the gradient sizes of different layers differently, whereas Adam always normalizes the gradient coordinatewise).

**Convolution**    A convolution weight tensor $W \in \mathbb{R}^{\text{fan\_out} \times \text{fan\_in} \times s_1 \times s_2}$ with kernel size $s_1 \times s_2$ can be thought of just as a $s_1 s_2 = \Theta(1)$-sized collection of fan\_out $\times$ fan\_in dense weights. Then all of our discussions above apply accordingly.

### L.3    Why Other Parametrizations Cannot Admit Hyperparameter Transfer

**Standard Parametrization (SP)**    SP doesn't work essentially because it leads to blow-up in the infinite-width limit.

1. For Adam with LR $\Theta(1)$, $\Delta W$ would have $\Theta(1)$ coordinates, causing preactivations to blow up like $\Theta(n)$ by Desideratum 1 and Table 14. We can avoid this blowup with LR $\Theta(1/n)$, but this induces a non-maximal feature learning limit, which, as we argue below, cannot transfer hyperparameters in all situations.
2. For SGD, the gradient of $\mathbb{R}^{n \times n}$ weight has $\Theta(1/\sqrt{n})$ coordinates, so $\Theta(1)$ learning rate would make preactivation scale like $\Theta(\sqrt{n})$ and hence blow up. If we use $\Theta(1/width)$ learning rate, then blow-up does not occur. However, this infinite-width limit is in the kernel regime [45] and thus does not allow HP transfer for the same reason that NTP below does not.

**Neural Tangent Parametrization (NTP)**    We have concrete examples, e.g. Word2Vec in [45], where the NTK limit has trivial performance — so HPs have no effect at all — vastly outperformed by finite-width networks — where HPs matter. More importantly, wider does not always do better in NTP, especially in tasks where feature learning is crucial [45]. So in the context of modern deep learning e.g. large language model pretraining, NTP (or SP with $\Theta(1/width)$ LR) does not make sense for wide neural networks.

**Other Parametrizations**    Recall the *Dynamical Dichotomy Theorem* proven in [45], which says that any nontrivial stable "natural parametrization" (formally, "*abc-parametrization*," [45]) either admits a feature learning limit or a kernel limit, but not both.

Our argument above against SP and NTP will also work against any parametrization inducing a kernel limit. Therefore, it remains to ask, can other *feature learning* parametrizations transfer HPs?

We argue no. As shown in [45], any other feature learning parametrization differs from $\mu$P essentially only in that some parameters are not updated maximally. By [45, Sec 6.4], in the infinite-width limit, such parameters can be thought of as being fixed at initialization. Therefore, in such infinite-width limits, the learning rate of such parameters becomes useless. Therefore, we cannot hope for the HP landscape of the limit to reflect the HP landscape of finite-width neural networks.

$\mu$P is the unique feature learning parametrization that updates all parameters maximally, so that the learning rate of each parameter plays approximately the same role in finite-width neural networks as in the infinite-width limit. Consequently, the HP landscape of the $\mu$P limit should reflect the HP landscape of finite-width neural networks.