# OpenReview forum: "Tuning Large Neural Networks via Zero-Shot Hyperparameter Transfer"
_NeurIPS.cc/2021/Conference — NeurIPS 2021 Poster_

### Official Review · Reviewer_Ty3V · 2021-07-16

**Rating:** 7
**Confidence:** 4

**Summary:**

Based on recently proposed Maximal Update Parametrization ($\mu$P), this paper finds that hyperparameters can be copied (i.e. transferred with zero cost) across model sizes. Therefore, it is possible to tune hyperparameters in a small model, and to apply hyperparameters in a large model.

**Limitations And Societal Impact:**

Limitations are discussed in the paper.

**Main Review:**

Although this paper claims it is Zero-Shot Hyperparameter Transfer, I would rather say that they find some scaling rules for hyperparameter tuning. This paper heavily depends on [40], and can be seen as an empirical validation of the theory proposed by [40]. I suggest that the authors weaken the claim that this is the first paper to explore zero-shot hyperparameter transfer, because learning rate scaling rules [31] / initialization rules (Kaiming initialization and Xavier initialization etc) all fall into this category of "zero-shot hyperparameter transfer".

Large models like GPT-3, T5 are not only difficult to train, but also difficult to tune. Even research institutions in large companies like google and facebook cannot afford grid-search tuning of hyperparameters for these large models. Therefore, **the demonstrations in this paper can have a huge impact for training large models**. It is thrilling to see that we can tune hyperparameters in a small Transformer and expect better performance in a large Transformer, which is also validated in ResNets. Based on the potential impact, I give a positive review. But to be honest, the technical novelty is limited, cause this paper is a direct application of the theory in [40].

I'd like to increase the score if my concerns are addressed:

(1) "this is the first paper to explore zero-shot hyperparameter transfer" is an over-claim. Please tone down the claim. Section 2 (Parametrization Matters: A Primer) talks about proper scaling of hyperparameters, not hyperparameter transfer. I think everyone can understand the importance of finding the proper rule of hyperparameters scaling with respect to model scaling. In my humble opinion, section 2 is uninformative. Maybe authors can move some content from the supplementary material to the main paper to replace section 2.

(2) why regularization cannot be transferred? Since the main goal of this paper is to verify hyperparameter scaling rules, intuitively there should also be a scaling rule for regularization factors.

**Time Spent Reviewing:**

10

---

> ### Author Response · Authors · 2021-08-09
> **Rebuttal**
>
> Thanks for your review.
>
> > the technical novelty is limited, cause this paper is a direct application of the theory in [40].
>
>
> Our work is not a direct application of [40]. Please see our main comment *Response to common concerns.*
>
> > I suggest that the authors weaken the claim that this is the first paper to explore zero-shot hyperparameter transfer, because learning rate scaling rules [31] / initialization rules (Kaiming initialization and Xavier initialization etc) all fall into this category of "zero-shot hyperparameter transfer".
>
> Thanks for your suggestion. We will tone down our claim.
>
>
> > Although this paper claims it is Zero-Shot Hyperparameter Transfer, I would rather say that they find some scaling rules for hyperparameter tuning.
>
> In our terminology, “K-shot hyperparameter transfer algorithm” means an algorithm that takes the optimal hyperparameter combination from a proxy model and returns a near-optimal hyperparameter combination for the target model, allowing up to K evaluations of the combination on the target model. So “zero-shot” means that we obtain near-optimal hyperparameters on the target model without any evaluation of the target model. In contrast, “scaling rule” is just some way of scaling hyperparameters recommended based on various reasons such as information propagation or for the purpose of hyperparameter transfer, without any claim of near-optimality. A hyperparameter transfer algorithm can use a scaling rule or use more clever algorithms like Hyperband (in the nonzero-shot setting). In our case, muTransfer is a zero-shot hyperparameter transfer algorithm, while muP is a scaling rule.
>
> > In my humble opinion, section 2 is uninformative. Maybe authors can move some content from the supplementary material to the main paper to replace section 2.
>
> We’d be happy to but Reviewer Z5cb asks us to expand section 2. We’d like to hear your opinion again after reconciling with Reviewer Z5db.
>
> > why regularization cannot be transferred? Since the main goal of this paper is to verify hyperparameter scaling rules, intuitively there should also be a scaling rule for regularization factors.
>
> As we say in line 178 to 179: “the amount of regularization naturally depends on both the model size and data size, so we should not expect transfer to work if the parametrization only depends on model size.” To be clear, we can still transfer weight decay etc using muP to the extent that they (possibly) help the training loss/accuracy, but the optimal weight decay etc for the test loss/accuracy for a smaller model is probably not optimal for a larger model on the same dataset because the latter can overfit more.
>
>
>
> If we have addressed your concerns, please consider raising your score. Thanks again!

---

> > ### Comment · Reviewer_Ty3V · 2021-08-26
> > **Comments on authors' Rebuttal**
> >
> > The authors' rebuttal partially addressed the novelty concern, but I think the over-claim stays about Hyperparameter Transfer. I hope that authors will tone down the claim and call their method a scaling rule in the future revision. That said, I choose to keep my score.

---

### Official Review · Reviewer_rfmG · 2021-07-16

**Rating:** 7
**Confidence:** 3

**Summary:**

The paper proposes an alternative method for tuning hyperparameters which is to tune hyperparameters on a small width model and transfer these sets of hyperparameters to the full network. The core idea is to utilize the maximal update parameterization so that optimization hyperparameters remain stable under model size changes. This enables a zero-shot transfer and makes the hyperparameter optimization process more efficient both in computation and memory. Empirically, the authors show that the careful hyperparameter selection on the small model with the proposed approach outperforms BERT-large on MNLI and OOP and also requires less total training costs.

**Limitations And Societal Impact:**

The authors adeptly address the potential negative social impact of their work.

**Main Review:**

Originality:
* The core principle of the method is based on Maximal Update Parametrization (\mu P) and the work is incremental to show hyperparameters such as learning rate are stable under model size changes.
* However, the idea of zero-shot hyperparameter transfer is interesting and introduces a different perspective in hyperparameter optimization.
* It is clear how this work differs from previous contributions and the related work is adequately cited.

Quality:
* The submission is technically sound and claims made in the paper are well supported. The authors carefully discussed the weakness and strengths of their proposed approach.

Clarify:
* The submission is clearly written and well-organized. It was a good read overall.

Significance:
* The results presented in the paper are important and interesting. I expect other researchers or practitioners likely to use the idea or build on them.
* One concern I have with the paper is the limited novelty. The core foundation of the paper is based on Maximal Update Parametrization (\mu P). However, the investigation of several hyperparameters' sensitivity on \mu P and experiments to analyze this behaviour is interesting and promising.
* Another concern I have is the limited generalizability (scalability). The proposed parameters cannot handle regularization hyperparameters such as weight decay and dropout. Since the modification of such regularization hyperparameters may change the optimization hyperparameters, in the end, to obtain the state-of-the-art results, I feel like hyperparmaeter search on the full model is inevitable. I believe that future works are necessary for the proposed model to be fully utilized in practice.

Minor Comments:
* In footnote 8, it should be "in fact" -- > "In fact".
* The use of \times instead of x would be helpful in section 6.

--------

I thank the authors for their response. I acknowledge that I read the authors' responses and other reviewers' comments. The authors addressed some of my concerns on the novelty of the proposed approach. Regarding the second issue, I understand that the proposed approach achieved good results in GPT-3 that doesn't require much regularization. However, for models that require careful tuning of regularization hyperparameters to obtain the state-of-the-art results (e.g. ImageNet classification), hyperparameter search on the full model is inevitable and the proposed method wouldn't be helpful. Given these reasons, I would like to remain at my current score.

**Time Spent Reviewing:**

5

---

> ### Author Response · Authors · 2021-08-09
> **Rebuttal**
>
> Thanks for your review.
>
>
> > The core principle of the method is based on Maximal Update Parametrization (\mu P) and the work is incremental to show hyperparameters such as learning rate are stable under model size changes.
>
>
> Our work is not a trivial application of [40]. Please see our main comment *Response to common concerns.*
>
>
> > Another concern I have is the limited generalizability (scalability). The proposed parameters cannot handle regularization hyperparameters such as weight decay and dropout. Since the modification of such regularization hyperparameters may change the optimization hyperparameters, in the end, to obtain the state-of-the-art results, I feel like hyperparmaeter search on the full model is inevitable. I believe that future works are necessary for the proposed model to be fully utilized in practice.
>
> Our work is already very useful for large scale pretraining like GPT3, where regularization is not needed and “hyperparmaeter search on the full model” is not only not inevitable but unaffordable. Please see our main comment *Response to common concerns.*
>
> If we have addressed your concerns, please consider raising your score. Thanks again!

---

### Official Review · Reviewer_5dwZ · 2021-07-16

**Rating:** 5
**Confidence:** 3

**Summary:**

The authors build on the recently developed Maximal Update Parameterisation
($\mu$P) to develop a framework for using the same hyperparameters on
models with variable numbers of hidden units or channels. The authors
demonstrate the value of this approach on large benchmark tasks, such as
IESLT14 De-En, WNT14 En-De and training BERT-large on wikitext-2.

**Limitations And Societal Impact:**

Comparisons in the experiment section to similar works or alternative
solutions for hyperparameter search are lacking. As noted in the main
review, the authors list that the methods they are comparing against
"diverged" in Tables 4,5 and 6. It would make more sense to compare against
a contemporary work focused on the same problem of hyperparameter transfer.
See the main review for suggestions.

**Main Review:**


## Originality

> Are the tasks or methods new? Is the work a novel
> combination of well-known techniques? (This can be valuable!) Is it clear
> how this work differs from previous contributions? Is related work
> adequately cited?

This is an application of the Maximal Update Parameterisation to the
problem of hyperparameter tuning. The paper provides theoretical and
extensive experimental exploration of this topic.

The experimental results showing transfer of hyperparameters regardless of
model size is the main original contribution of this paper.

## Quality

> Is the submission technically sound? Are claims well supported
> (e.g., by theoretical analysis or experimental results)? Are the methods
> used appropriate? Is this a complete piece of work or work in progress? Are
> the authors careful and honest about evaluating both the strengths and
> weaknesses of their work?

Experimental results are concrete and described in great detail. The
theory described in the paper describes the Maximal Update Parameterisation
and the background for it's development. The aim of this parameterisation
is to be invariant to hyperparameters so it is natural that it would be
useful for this task.

Tables 4, 5 and 6 list experimental results where the training of models
that are being compared against diverged. This does not give a fair
comparison to existing work. In practice, practitioners would try to tune
the model hyperparameters to the new setting using hyperparameter tuning
techniques or personal expertise. A standardised way to train the models so
that they do not diverge would be a fairer comparison.

A natural choice for this would be to automate the traditional method of
tuning hyperparameters on a subset of the full dataset or a single
minibatch before running it on the full dataset. Random search over
hyperparameters could provide a good baseline using this method to compare
against.

The experiments presented expand upon the cases where this method of
hyperparameter transfer performs well but fail to explore where this model
fails. For example, it is mentioned that the method does not transfer to
architectures using different regularisation methods but this is not
explored in experiment.

It is not clear how this model behaves while varying the hyperparameters
that can be transferred across, as listed in Table 1. The authors do not
explore the regions where the maximal update parameterisation breaks down.
In other words, how much does the underlying architecture have to change
before training with hyperparameters chosen on a small network breaks down?

## Clarity

> Is the submission clearly written? Is it well organized? (If not, please
> make constructive suggestions for improving its clarity.) Does it
> adequately inform the reader? (Note that a superbly written paper provides
> enough information for an expert reader to reproduce its results.)

The work very clearly states its strengths and immediately explains why the
parameterisation is ideal for this task. Figure 1 is well placed to explain
this to the reader and communicates the main thesis of the paper clearly.

The paper editorializes unnecessarily. For example, at the start of Section
3 and Section 5.1, the authors make a statement about community opinion
without citation. Footnotes generally also express the author's opinions.

In the introduction a sequence of numerical bullet points are presented in
a paragraph instead of as a list. It would be clearer as a list.

## Significance

> Are the results important? Are others (researchers or practitioners) likely
> to use the ideas or build on them? Does the submission address a difficult
> task in a better way than previous work?  Does it advance the state of the
> art in a demonstrable way? Does it provide unique data, unique conclusions
> about existing data, or a unique theoretical or experimental approach?

Parameterisation in deep learning is important. This work demonstrates a
valuable application of a specific parameterisation and how this can
massively improve contemporary deep learning practice.

The experiments provided are extensive and could provide a useful reference
to researchers who would like to use the proposed parameterisation.

One detriment to the significance of this work is that it did not introduce
the Maximal Update Parameterisation. The experiments in this work follow
directly from that development without extra development work. The major
contribution of this work is the experimental results demonstrating that
this parameterisation still works on large deep learning problems.

**Time Spent Reviewing:**

3

---

> ### Author Response · Authors · 2021-08-09
> **Rebuttal**
>
> Thank you for your review.
>
> > The aim of this parameterisation is to be invariant to hyperparameters so it is natural that it would be useful for this task.
>
> Our work is not a trivial application of [40] and a priori it’s not clear that muP is useful for hyperparameter transfer. Please see our main comment *Response to common concerns.*
>
> > Tables 4, 5 and 6 list experimental results where the training of models that are being compared against diverged. This does not give a fair comparison to existing work. In practice, practitioners would try to tune the model hyperparameters to the new setting using hyperparameter tuning techniques or personal expertise. A standardised way to train the models so that they do not diverge would be a fairer comparison. A natural choice for this would be to automate the traditional method of tuning hyperparameters on a subset of the full dataset or a single minibatch before running it on the full dataset. Random search over hyperparameters could provide a good baseline using this method to compare against.
>
> There seems to be some misunderstanding.
> We already tune directly on the target model using random search in the setup denoted as “Tuning on 1x” in Tables 4 and 5. We did not tune BERT directly because the tuning cost is too expensive. The diverged models are from naive transfer, i.e. copy the hyperparameters from the small model to the large model in SP instead of muP (setup is denoted as “Naive transfer from 0.25x” in Tables 4 and 5 and “Naive transfer” in Table 6). This is what people would do currently if they would like to “zero-shot transfer” and we are showing it is not feasible at all compared to muTransfer.
>
> For the BERT experiments we also already tuned the proxy model on a smaller subset of the dataset as the reviewer suggests.
>
> > The experiments presented expand upon the cases where this method of hyperparameter transfer performs well but fail to explore where this model fails. For example, it is mentioned that the method does not transfer to architectures using different regularisation methods but this is not explored in experiment.
>
> We will add experiments exploring the limits of our transfer and for regularization.
>
> > The paper editorializes unnecessarily.
>
> We will improve our paper according to your suggestion
>
> > One detriment to the significance of this work is that it did not introduce the Maximal Update Parameterisation. The experiments in this work follow directly from that development without extra development work.
>
> Our work is not a trivial application of [40]. For example, we introduced muP for Adam in this paper. Please see our main comment *Response to common concerns.*
>
> > Comparisons in the experiment section to similar works or alternative solutions for hyperparameter search are lacking. As noted in the main review, the authors list that the methods they are comparing against "diverged" in Tables 4,5 and 6. It would make more sense to compare against a contemporary work focused on the same problem of hyperparameter transfer.
>
> We believe our response above should have addressed your concern here.
>
>
>
> If we have addressed your concerns, please consider raising your score. Thanks again!

---

### Official Review · Reviewer_Z5cb · 2021-07-20

**Rating:** 6
**Confidence:** 4

**Summary:**

Summary:

This paper makes the observation that when initializing neural network with the *maximal

update parametrization* technique, certain hyperparameters such as learning rate and learning rate schedule are transferrable across different scales of the network (e.g. width of each layer). This insight enables to authors to tune hyperpameters using a small network and directly transfer the tuned results to much larger networks, saving the cost of tuning larger networks. Although such tuning-and-transfer approach is probably already a standard practice, the authors point out that such practice is not optimal if we initialize neural networks using the standard technique and using the proposed method generally leads to much better transfer result.

I find this work to be interesting and see the potential of this submission to inspire future work. That being said, I want to point out that the *maximal update parametrization* technique is already published in a previous work and this submission should be viewed more as an add-on to this prior work. Practically speaking, I should also note that not all hyperparameters are transferrable as pointed out by the authors candidly. Certain hyperpameters such as regularization strength are not transferrable across scale. Such hyperparameters may have complex interactions with other hyperparameters and therefore make the proposed method less useful. Given these two reasons, I am leaning toward accept but have some reservation. I detail my other concerns in the following section.

**Ethical Concerns:**

I don't have ethical concerns with this work.

**Ethics Review Area:**

["I don’t know"]

**Limitations And Societal Impact:**

I don't see negative societal impact from this work.

**Main Review:**

Pros:

- the submission is novel and aims at solving a practical problem
- the paper is well-written

Cons/Questions:

- the submission is largely built upon the prior work "maximal update parametrization"
- I wish the author spent more time on explaining the intuition of the finding, namely in section 2. In line 91, the authors mention that the hyperparameters are similar to "c". However, I fail to see this connection. I am wondering whether is it reasonable to think of the  test-set perforance as a bounded continuous function of the hyperparamter. I would suggest the authors to expand on this analogy and also try to provide more intuition of this finding.
- I want to double check my understanding of the experimental details.
    - In Table 6, how is the total speed up ratio calculated? Why is it different from model speedup?
    - How does such direct transfer of hyperparameter compare to tuning the hyperparameter with a short amount of time? In my experience, the training loss decreases much slower toward the second half of pretraining and we should be able to tell the better hyperparameters early on. If the authors have saved all the intermediate checkpoints for BERT pretraining, this experiment should be easy to add.
- nitpick: not sure if the terminology "zero-shot hyperparameter transfer' makes sense to me because it is not clear what "few-shot" or "many-shot" hyperparameter transfer would be.

**Time Spent Reviewing:**

6

---

> ### Author Response · Authors · 2021-08-09
> **Response**
>
> Thanks for your review.
>
> > This paper makes the observation that when initializing neural network with the *maximal
> update parametrization* technique,
>
> Note that we change not only the initialization but also the multipliers and learning rates (or more generally, the model parametrization). For Adam, the learning rate scaling is crucial. For this reason, we emphasize the term “parametrization” as opposed to “initialization” in our paper.
>
> > Although such tuning-and-transfer approach is probably already a standard practice, the authors point out that such practice is not optimal
>
> It's not standard practice to the best of our knowledge and we'd love to see evidence suggesting otherwise. Also as shown in our expts, a naive attempt at hyperparameter transfer isn’t simply “suboptimal” -- it completely fails once we increase the scale across which we transfer. It is still not expected to be performant even given some empirical scaling rules as shown in the example of figure 1, as these rules assume a flawed parametrization.
>
> >  I want to point out that the maximal update parametrization technique is already published in a previous work and this submission should be viewed more as an add-on to this prior work.
>
> Our work is not a trivial application of [40]. Please see our main comment *response to common concerns*.
>
> > Certain hyperpameters such as regularization strength are not transferrable across scale. Such hyperparameters may have complex interactions with other hyperparameters and therefore make the proposed method less useful.
>
> Even without transferring regularization, our work is already very useful for large scale pretraining. Please see our main comment *response to common concerns*.
>
> > I wish the author spent more time on explaining the intuition of the finding, namely in section 2.
>
> Immediately after our submission, we in fact revised our draft by spending more time explaining the intuition by working through the theory of simple cases and also by showing empirically how logits, attention logits, and different activations can blow up in a SP transformer.
>
> > In line 91, the authors mention that the hyperparameters are similar to "c". However, I fail to see this connection.
>
> Each weight gradient has the form $g x^T$ where $g$ is the preactivation gradient coming back and $x$ is the incoming activation. Thus after one step of SGD (with batch size 1), we have $W_{t+1} = W_t + \eta g x^T$. In the next forward pass, suppose the incoming activation is now $x’$, so the preactivation $W_{t+1} x’ = W_t x’ + \eta g_t (x^T x’) = W_t x’ + n \eta g_t \frac{x^T x’}{n}$. As width (dimension of $x$ and $x’$) tends to infinity, $\frac{x^T x’}{n}$ will converge to some deterministic scalar, like how in our example, $(x_1 + \cdots x_n)/\sqrt{n}$ tends to a Gaussian via Central Limit Theorem. In both cases, $\eta$ or $c$ acts as a multiplier to this weighted sum.
>
> > I am wondering whether is it reasonable to think of the test-set perforance as a bounded continuous function of the hyperparamter.
>
> In our experience, the training and test set accuracies and losses are both continuous in hyperparameters, as exemplified by all of our plots. The accuracies are obviously bounded, and the losses are in practice bounded above by some large number and below by 0.
>
> In fact, we only assume $f$ as “bounded continuous” as an example so that Central Limit Theorem goes through easily. In reality, as long as the network nonlinearities have polynomially bounded 2nd derivatives and the loss function has a continuous derivative, then by the Tensor Programs technique [39], the training and test set losses are guaranteed to converge as width tends to infinity.
>
> > I would suggest the authors to expand on this analogy and also try to provide more intuition of this finding.
>
> We would like to but Reviewer Ty3V is advocating for the removal of section 2. We’d like to hear your recommendation after reconciling with Reviewer Ty3V.
>
> > In Table 6, how is the total speed up ratio calculated? Why is it different from model speedup?
>
> As stated in line 267 - 268, when we tune, we only train for 10^5 steps, but when we evaluate the target model, we train for the full 10^6 steps (so we are transferring across time steps in addition to width and depth). Model speedup only accounts for transferring across width and depth, while total speedup accounts for time steps as well.
>
> >  How does such direct transfer of hyperparameter compare to tuning the hyperparameter with a short amount of time? In my experience, the training loss decreases much slower toward the second half of pretraining and we should be able to tell the better hyperparameters early on. If the authors have saved all the intermediate checkpoints for BERT pretraining, this experiment should be easy to add.
>
> We are already doing that. All transfer results in Table 6 include transfer across training time.
>
> > not sure if the terminology "zero-shot hyperparameter transfer' makes sense to me because it is not clear what "few-shot" or "many-shot" hyperparameter transfer would be.
>
> We will state the meaning more formally: “K-shot hyperparameter transfer” means an algorithm that takes the optimal hyperparameter combination from a proxy model and returns a *near-optimal* hyperparameter combination for the target model, allowing up to K evaluations of hyperparameter combinations on the target model. So “zero-shot” means that we obtain near-optimal hyperparameters on the target model without any evaluation of the target model. This is important, as Reviewer Ty3V mentions, because for very large models like T5 or GPT3, one can afford at most one training run so there is no budget to do any direct hyperparameter tuning at all on the target model.
>
> If we have addressed your concerns, please consider raising your score. Thanks again!

---

### Official Review · Reviewer_dYRN · 2021-07-25

**Rating:** 6
**Confidence:** 4

**Summary:**

The authors demonstrate the effectiveness of zero-shot transfer optimal hyperparameters from small models to large models within the same model family. They verify their method on machine translation and large language model pre-training.

**Limitations And Societal Impact:**

Yes

**Main Review:**

Pro:
The paper is well written and easy to follow. They first parameterize the target model in Maximal Update Parameterization, then tune the smaller version of the target model and copy the optimal hyperparameters to the large model.

- The authors acknowledge the limitation of \muTransfer clearly: It applies to pretraining the architectures in the same family to the same data, but cannot transfer regularization parameters or for fine-tuning for other datasets.

Cons:

- The paper is mainly based on the Maximal Update Parameterization, which was proposed in [40]. However, the authors assume readers are familiar with the work.

- The authors find that for language modeling on Transformers, hyperparameters generally transfer across scale dimensions if some minimum width, depth, batch size, sequence length, and training steps are met. Not too much explanation is provided here. How are these specific conditions obtained? Does this indicate the transability of the method is only applicable in certain conditions.

- The method enables HP transfer across width, however, a stronger model in the model family usually comes from increased depth, which also contributed the most to the latency of the model with the number of weights being the same compared to a wider model. It is not clear whether the method can generalize to depth.

- The authors mostly take shallow networks for illustrations or experiments to show the instability of optimal HPs across width, such as Figure 3 & 4. However, it is not clear whether it still holds when the network depth increases.

- In Fig 9, the variation of optimal HPs for ResNets with SP on CIFAR-10 is not quite evident. Also for large datasets such as ImageNet, the figures of instability of SP are not provided and the improvement of \muTransfer over SP is not that significant (0.41%). What is the gain of HPO in comparison with the default HPs, such as LR=0.1, momentum 0.9 in the original paper? It is not clear to me why some wired HPs used in the ResNets experiment in D.1.2, such as 2.048, 3.05, 0.875.


**Time Spent Reviewing:**

6

---

> ### Author Response · Authors · 2021-08-09
> **Response**
>
> Thank you for your review.
>
> > The paper is mainly based on the Maximal Update Parameterization, which was proposed in [40]. However, the authors assume readers are familiar with the work.
>
> We aim for this paper to be self-contained, and the reader should not need to know [40]. Could you point out where you think the knowledge of [40] was required in our paper?
>
> Also see our main comment *Response to common concerns*.
>
> > The authors find that for language modeling on Transformers, hyperparameters generally transfer across scale dimensions if some minimum width, depth, batch size, sequence length, and training steps are met. Not too much explanation is provided here. How are these specific conditions obtained? Does this indicate the transability of the method is only applicable in certain conditions.
>
> We estimated them from figure 4 and figure 10 as rules of thumb, and verified that these rules yield effective results in the experiments of section 6. We definitely do not expect muTransfer to work well when the width of the proxy model is too small, because the hyperparameter landscape would be too noisy and its optimum is possibly very biased away from the optimum of wide models. The latter concern also holds for depth, batch size, sequence length, and training steps.
>
> > The method enables HP transfer across width, however, a stronger model in the model family usually comes from increased depth, which also contributed the most to the latency of the model with the number of weights being the same compared to a wider model. It is not clear whether the method can generalize to depth.
>
> We empirically showed that we can muTransfer across depth in Fig 4. In our BERT experiment (table 6), we are also transferring across depth.
>
>
> > The authors mostly take shallow networks for illustrations or experiments to show the instability of optimal HPs across width, such as Figure 3 & 4. However, it is not clear whether it still holds when the network depth increases.
>
> Figure 13 shows the LR instability of an IWSLT transformer in standard parametrization. This transformer has 6 encoder and 6 decoder layers (so 12 layers in total).
>
> > In Fig 9, the variation of optimal HPs for ResNets with SP on CIFAR-10 is not quite evident.
>
> For ResNet, we believe the optimal hyperparameters may not change much in the range of width displayed due to several reasons: 1) SP is much closer to muP for SGD than for Adam. 2) The use of convolution means that more memory is needed for storing the activation pixel values, so we cannot go as wide as in a transformer on a single GPU in these experiments.
>
> Nevertheless, the key takeaway of Fig 9 is that the widest SP ResNet (with learning rate tuned) performs noticeably worse than the widest muP ResNet (with learning rate tuned), which is consistent with Fig 1.
>
> > Also for large datasets such as ImageNet, the figures of instability of SP are not provided and the improvement of \muTransfer over SP is not that significant (0.41%).
>
> It is very expensive to obtain the learning rate instability plot for ImageNet, and we do not believe it’s reasonable to ask us of this.
>
> For the same reasons above --- that 1) SP is much closer to muP for SGD than for Adam and 2) we cannot go as wide in a ResNet as in a Transformer due to memory --- we will see much more impressive gains on Transformers than on a ResNet due to muTransfer. Nevertheless, the takeaway of this experiment is that muP and muTransfer is always strictly better than SP and naive transfer, so it never hurts to reparametrize one’s model in muP.
>
> > It is not clear to me why some wired HPs used in the ResNets experiment in D.1.2, such as 2.048, 3.05, 0.875.
>
> Our repo is based on NVIDIA’s ConvNet repo, and these hyperparameters are the default (see https://github.com/NVIDIA/DeepLearningExamples/blob/f0ef8493eb077dccfa32fcecc90df869699a4365/PyTorch/Classification/ConvNets/resnet50v1.5/training/AMP/DGX1_RN50_AMP_50E.sh)
>
>
> If we have addressed your concerns, please consider raising your score. Thanks again!

---

### Author Response · Authors · 2021-08-09
**Response to common concerns**

Thanks to all reviewers for your time.

Here we respond to common concerns among all reviewers.

> Lack of novelty because MUP was derived in prior work

We believe there is some misunderstanding of our work and prior work [40] here.

1) The derivation of MUP for Adam and Transformers is a nontrivial contribution of this work (see Appendix G), which is what is used in our main empirical results. [40] only dealt with SGD and MLP. The derivation for Adam needs to consider *nonlinear tensor product* matrices arising from Adam updates and how they correlate with neural network activations and gradients. This was not considered at all in [40].

2) Several reviewers thought MUP was designed by [40] for hyperparameter transfer. But that’s not true. It was derived as the unique parametrization yielding the maximal feature learning infinite-width limit, in contrast to the popular NTK limit that does not learn features --- there was no mention anywhere of hyperparameter transfer or “hyperparameter invariance” in [40].

3) A priori, it’s not entirely clear why having the maximal feature learning limit should allow hyperparameters to transfer; for example, why would SP or NTK parametrization fail at transferring hyperparameters? A contribution of this work is to argue that MUP is the unique parametrization allowing such a transfer. This is summarized in line 558 to 568 and is fully presented in Appendix G.3.

4) More generally, [40] is targeted at a very theoretical audience. Our work, and in particular Sections 2, 3, 4, G, seeks to explain the core insights in a more accessible way and to make them useful for empirical researchers and engineers, so that for new architectures in the future, anyone can derive the right parametrization themselves.

Therefore, our work is not just a trivial application of [40]. In the next revision, we will emphasize the above contributions more clearly in the main text.

> muTransfer is not useful because regularization cannot be transferred

We believe muTransfer has the largest impact on large scale pretraining like in T5 or GPT3, where one can only afford a single training run. Here the data is in fact still much larger than the model size, so regularization is typically not used. So not being able to transfer regularization does not matter here.

In addition, our work shows that incorrect parametrization can lead to severe training convergence issues when scaling up, and muP and muTransfer can fix them even when we cannot transfer regularization.

---

### Decision · Program_Chairs · 2021-09-27

**Decision:**

Accept (Poster)

**Comment:**

This paper proposes an approach for "zero-shot transfer" of optimal hyperparameters from small models to large models, which enables efficient tuning of large models such as GPT-3. Such a study on efficient tuning of large models is highly anticipated by our community, and may inspire future research in this area. Reviewers generally recommended acceptance (4 out of 5), acknowledging the significance and experimentation of this work. They, with one referee still below the bar, pointed out several important concerns. Authors shall revise their paper to: 1. Elaborate the novelty by connecting to the Maximal Update Parameterization [40], highlighting clearly which part is due to [40] and which part is novel in this paper; 2. Town down the claim of "zero-shot" hyperparameter transfer, because hyperparameters of the regularization and the optimizers cannot be transferred, and the hyperparameter transfer can only be applied within the same model family; 3. Document clearly in what cases their approach is applicable, e.g. which hyperparameters can be transferred and which ones cannot, and how practitioners can deal with the difficult ones -- simply bypassing difficult cases is not good, because it will make scope of the approach quite narrow. I would stress that the author rebuttal did not respond to all the concerns, so the acceptance is conditioned on the authors' revision, in which at least the meta reviews must be well addressed.